# Spatially patterned excitatory neuron subtypes and projections of the claustrum

**Sarah R Erwin**[1†], **Brianna N Bristow**[1†], **Kaitlin E Sullivan**[1], **Rennie M Kendrick**[1], **Brian Marriott**[2], **Lihua Wang**[3], **Jody Clements**[3], **Andrew L Lemire**[3], **Jesse Jackson**[2,4], **Mark S Cembrowski**[1,3,5,6]*

[1]Department of Cellular and Physiological Sciences, Life Sciences Institute, University of British Columbia, Vancouver, Canada; [2]Neuroscience and Mental Health Institute, University of Alberta, Edmonton, Canada; [3]Janelia Research Campus, Howard Hughes Medical Institute, Ashburn, United States; [4]Department of Physiology, University of Alberta, Edmonton, Canada; [5]Djavad Mowafaghian Centre for Brain Health, University of British Columbia, Vancouver, Canada; [6]School of Biomedical Engineering, University of British Columbia, Vancouver, Canada

**Abstract** The claustrum is a functionally and structurally complex brain region, whose very spatial extent remains debated. Histochemical-based approaches typically treat the claustrum as a relatively narrow anatomical region that primarily projects to the neocortex, whereas circuit-based approaches can suggest a broader claustrum region containing projections to the neocortex and other regions. Here, in the mouse, we took a bottom-up and cell-type-specific approach to complement and possibly unite these seemingly disparate conclusions. Using single-cell RNA-sequencing, we found that the claustrum comprises two excitatory neuron subtypes that are differentiable from the surrounding cortex. Multicolor retrograde tracing in conjunction with 12-channel multiplexed in situ hybridization revealed a core-shell spatial arrangement of these subtypes, as well as differential downstream targets. Thus, the claustrum comprises excitatory neuron subtypes with distinct molecular and projection properties, whose spatial patterns reflect the narrower and broader claustral extents debated in previous research. This subtype-specific heterogeneity likely shapes the functional complexity of the claustrum.

*For correspondence:
mark.cembrowski@ubc.ca

†These authors contributed equally to this work

**Competing interest:** The authors declare that no competing interests exist.

## Introduction

The claustrum has been implicated in a variety of functions and behaviors, including attention (*Atlan et al., 2018*; *Goll et al., 2015*; *Smith et al., 2019*), impulsivity (*Liu et al., 2019*), sleep (*Narikiyo et al., 2020*; *Norimoto et al., 2020*; *Renouard et al., 2015*), and the integration of information to support consciousness (*Crick and Koch, 2005*; *Smythies et al., 2012*). To determine the mechanistic contributions of the claustrum to these putative functions, it is essential to understand both the intrinsic organization of claustrum neurons, as well as how this organization relates to connectivity and function (*Edelstein and Denaro, 2004*). However, such an interpretation is challenged by the fact that even the precise anatomical boundaries of the claustrum are a matter of debate (*Dillingham et al., 2019*; *Smith et al., 2019*).

Here, utilizing a multimodal cell typing approach, we sought to understand the extent of heterogeneity within the excitatory claustrum neuron population and relate this to local boundaries and long-range projections. Beginning with single-cell RNA sequencing, we identified two discrete populations of excitatory claustral neurons. To map the topography of these populations, we used multiplexed single-molecule fluorescent in situ hybridization, revealing core and shell claustral neuron subtypes that were transcriptionally distinguishable relative to surrounding cortical neurons. Combining this

with multicolor retrograde tracing, we revealed a spatial organization of distinct cortical-projecting claustral populations that mapped onto the identified core and shell subtypes. This work demonstrates that the claustrum consists of heterogeneous populations of excitatory neurons that are topographically organized and project to functionally dissociable cortical regions, suggesting subtype-specific functionality of excitatory claustral neurons. To facilitate future research analyzing claustral cell-type-specific structure and function, data and analysis tools from this study are available via our interactive web portal (http://scrnaseq.janelia.org/claustrum).

## Results

### scRNA-seq reveals discrete excitatory neuron subtypes within the claustrum

We began by using single-cell RNA sequencing (scRNA-seq) to understand the transcriptomic organization of the claustrum. From claustral microdissections from four mice, we manually harvested 1112 cells based on a combination of unbiased blind selection of cells and selection of specific labeled projections (to either the lateral entorhinal cortex [LEC] or retrosplenial cortex [RSC]; see Materials and methods). After library preparation, sequencing, and filtering, we retained a total of 1011 excitatory neurons for analysis (n = 478 cells blindly selected; n = 286 and 247 cells projecting to the LEC and RSC, respectively).

We initially examined this dataset agnostic to any projection-specific information. Combining UMAP nonlinear dimensionality reduction (*McInnes et al., 2018*) with Louvain graph-based clustering (*Stuart et al., 2019*) revealed that cells broadly conformed to three transcriptomically separated clusters (*Figure 1A*; also seen in t-SNE: *Figure 1—figure supplement 1A*). These clusters were all associated with expression of excitatory neuronal markers (*Figure 1B*) and were found across the anterior-posterior axis and across animals (*Figure 1—figure supplement 1B,C*). In seeking to assign transcriptomic phenotypes to these cells, we noted one cluster ('Cluster 1') was enriched for the claustrum marker gene *Synpr* (*Binks et al., 2019*; *Wang et al., 2017*). This cluster and a second cluster ('Cluster 2') exhibited enriched expression of other claustrum marker genes (e.g., *Gnb4*) relative to the third cluster ('Cluster 3') (*Figure 1C*), with Cluster 2 also showing uniquely expressed marker genes (e.g., *Slc30a3*; *Figure 1D*). Conversely, Cluster 3 was enriched for markers of excitatory cortical populations (e.g., the layer 6b marker *Ctgf* and the layer 6a marker *Sla*) (*Tasic et al., 2016*; *Tasic et al., 2018*), suggesting a cortical phenotype (*Figure 1C*). Each cluster also exhibited enriched expression of many other genes associated with neuronal function (*Figure 1E*), suggesting structural and functional heterogeneity between these three clusters (for full lists of differentially expressed genes, see *Supplementary files 1–3*).

### Comparison to other scRNA-seq data

To understand our results in the context of other published scRNA-seq data, we next integrated our dataset with existing large-scale datasets that potentially included the claustrum (*Saunders et al., 2018*; *Zeisel et al., 2018*). Consistent with the three clusters identified within our dataset, our dataset largely conformed to three distinct locations within the broader cell-type landscape when incorporating published data (*Figure 1—figure supplement 2A,B*). In particular, Cluster 1 cells occupied an isolated group of *Synpr*-expressing cells, whereas Cluster 2 cells coarsely occupied a distinct location nearby other datasets, but were also enriched for specific marker genes like *Nnat* (*Figure 1—figure supplement 2C*). In agreement with Cluster 2 being non-cortical, these *Nnat*-expressing Cluster 2 cells were also depleted in *Pcp4* expression (*Figure 1—figure supplement 3*), a gene strongly expressed in deep cortical layers (but with exception of layer 6 intratelencephalic excitatory neurons: *Watakabe et al., 2012*; *Figure 1—figure supplement 4*). In collection, this work shows that our scRNA-seq data recapitulates previously described cell types and further suggests new marker genes and specializations that may have been underresolved in previous studies.

### Two types of excitatory claustral neurons exist in a core-shell arrangement

As the spatial cell-type-specific organization of the claustrum remains uncertain, we next sought to map our scRNA-seq-identified cell types in a spatial context. To do this, we used multiplexed

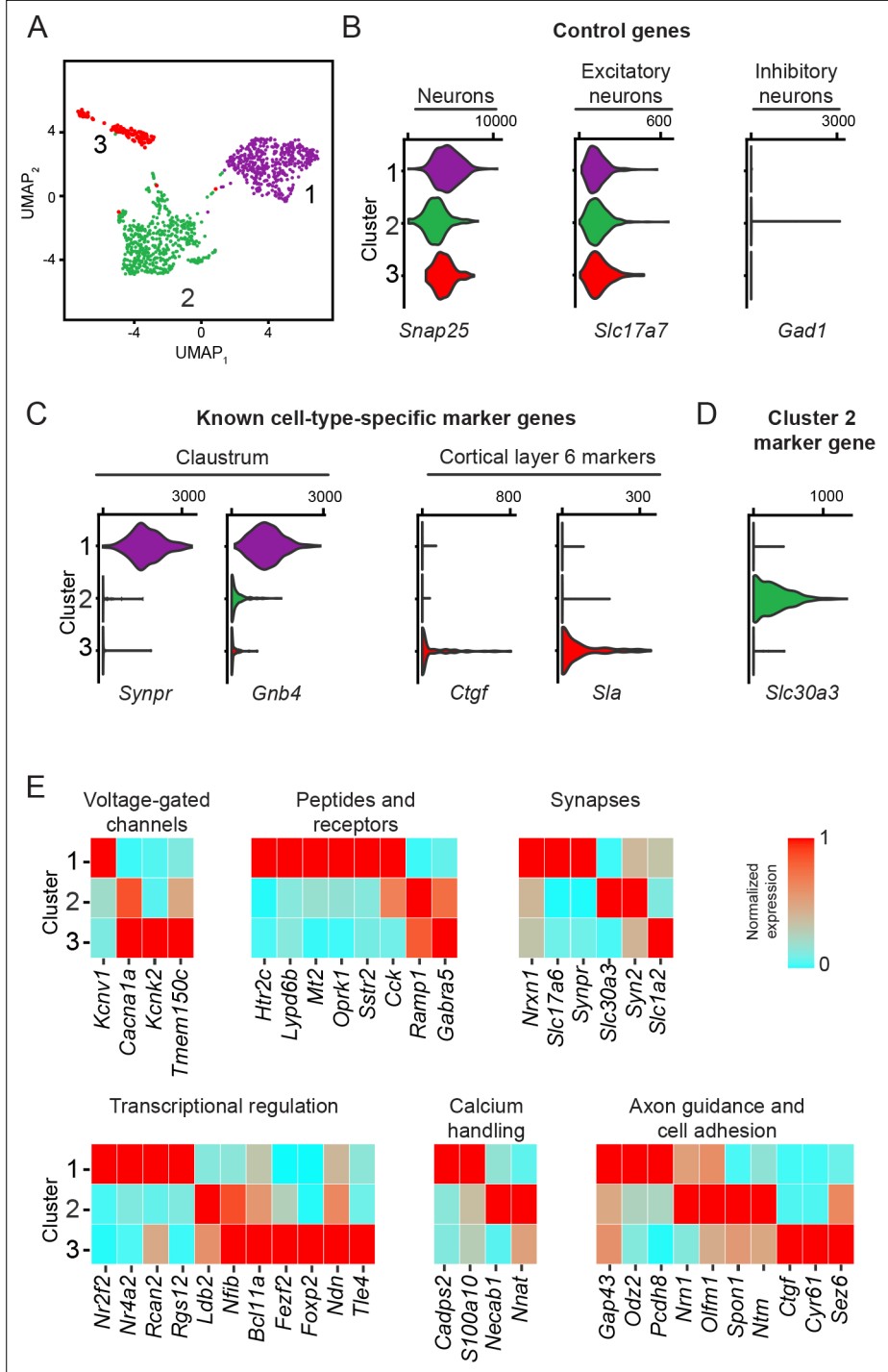

**Figure 1.** Excitatory claustrum-occupying cells are separable into discrete transcriptomic populations. (**A**) UMAP dimensionality reduction of single-cell transcriptomes. Points denote individual cells, with coloring denoting cluster identity obtained by graph-based clustering. (**B**) Violin plots illustrating expression of control marker genes, with accompanying values denoting normalized and log-transformed count value associated with right tick mark. (**C**) As in (**B**), but for known marker genes of claustrum neurons and layer 6 cortical neurons. (**D**) As in (**B**), but for the novel Cluster 2 marker gene *Slc30a3*. (**E**) Heatmap illustrating expression of genes associated with neuronal functionality that are enriched or depleted in a cell-type-specific fashion.

The online version of this article includes the following figure supplement(s) for figure 1:

**Figure supplement 1.** Consistency and reproducibility of scRNA-seq data.

*Figure 1 continued on next page*

*Figure 1 continued*

**Figure supplement 2.** Comparison of new and published scRNA-seq datasets.

**Figure supplement 3.** *Nnat* and *Pcp4* differentiate similar cells across datasets.

**Figure supplement 4.** *Pcp4* and *Nnat* differentiate deep cortical layers from claustrum shell.

fluorescent in situ hybridization (mFISH) (*Wang et al., 2012*), allowing us to map 12 RNA targets at a single-molecule and single-cell resolution (*Sullivan et al., 2020*). We selected genes that allowed us to grossly identify excitatory neurons (*Slc17a7, Slc17a6*), cortical markers (*Ctgf, Pcp4*), classical claustrum markers (*Synpr, Lxn, Gnb4*), and putative subtype-specific markers from our scRNA-seq dataset (*Cdh9, Slc30a3, Gfra1, Nnat, Spon1*) (overview of genes in scRNA-seq: *Figure 2—figure supplement 1*).

We used mFISH to spatially register expression of these 12 genes across anterior, intermediate, and posterior sections of the claustrum (*Figure 2A*; expansions: *Figure 2—figure supplement 2*; *Video 1*; n = 18,957 excitatory neurons from n = 5 animals analyzed). In doing so, we identified a claustrum population with a relatively central core-like location that exhibited expression of *Synpr*, and a surrounding shell-like population exhibiting expression of *Nnat* (*Figure 2B–D*). This organization was present across the anterior-posterior axis (*Figure 2—figure supplement 3*) as well as across animals (*Figure 2—figure supplement 4*), and recapitulated gene-expression properties predicted from scRNA-seq (*Figure 2—figure supplement 5*). Adjacent to these populations were other neuronal subtypes enriched for markers of cortical cells, including a cluster with spatial and transcriptional properties of deep layer 6 cells (i.e., a *Ctgf*-expressing cluster in the deepest cortical layer). Collectively, these results illustrated that claustrum excitatory neuron subtypes are transcriptionally distinct from neighboring cortical neurons and form a core-shell spatial organization.

## Claustrum excitatory subpopulations co-vary with projection target

Does this differential marker gene expression and spatial patterning correspond to distinct claustral projections? To answer this question, we next considered projections to the RSC and LEC, two claustral projections that exhibit minimal overlap (two-color retrograde viral injections: *Figure 3A*; see also *Marriott et al., 2020*). We first examined our scRNA-seq dataset with respect to projection targets, where a subset of RSC- and LEC-projecting cells were specifically targeted by retrograde labeling and manual harvesting (*Figure 3B*). Strikingly, 85 % (204/241) of RSC-projecting claustrum cells mapped onto the *Synpr*-expressing class, whereas 84 % (238/282) of LEC-projecting claustrum cells mapped onto the *Nnat*-expressing class (*Figure 3C*). Similarly, applying mFISH to retrograde-labeled cells provided complementary evidence that *Synpr* and *Nnat* were respectively enriched in RSC-projecting and LEC-projecting cells (representative section: *Figure 3D–H*; all projection cells: *Figure 3I*, *Figure 3—figure supplement 1*), and illustrated that RSC- and LEC-projecting cells were enriched in distinct claustral subtypes (216/259 = 83% of RSC-projecting claustral cells were found in core cluster and 276/324 = 85% of LEC-projecting claustral cells were found in shell cluster, n = 4 and n = 2 animals, respectively, *Figure 3J*). Thus, distinct excitatory claustrum projection neurons were coherently separable by marker genes, local spatial organization, and long-range projection targets.

## Discussion

A variety of different approaches have previously been used to establish the anatomical definitions of the claustrum. Marker-based approaches using individual genes such as *Lxn, Gnb4*, and *Slc17a6* coarsely demarcate the boundaries of the adult mouse claustrum (*Fodoulian et al., 2020*; *Kitanishi and Matsuo, 2017*; *Mathur et al., 2009*; *Peng et al., 2020*; *Wang et al., 2017*; *Watakabe et al., 2012*), but it is unclear if these different genes all converge upon a monolithic cellular population or embody different claustrum subtypes (and potentially include phenotypically cortical cells: *Bruguier et al., 2020*; *Molnár et al., 2020*; *Puelles et al., 2016*). Retrograde tracing from the cortex has been useful for identifying claustrum projection neurons (*Marriott et al., 2020*; *Minciacchi et al., 1985*; *Watson et al., 2017*; *Zingg et al., 2018*), but similarly it remains unknown whether projection-labeled claustrum cells are intrinsically homogeneous.

Our approach here, integrating transcriptomic and circuit-level approaches, identified two claustrum cell subtypes that are molecularly distinguishable from surrounding cortex (*Figures 1 and 2*) and

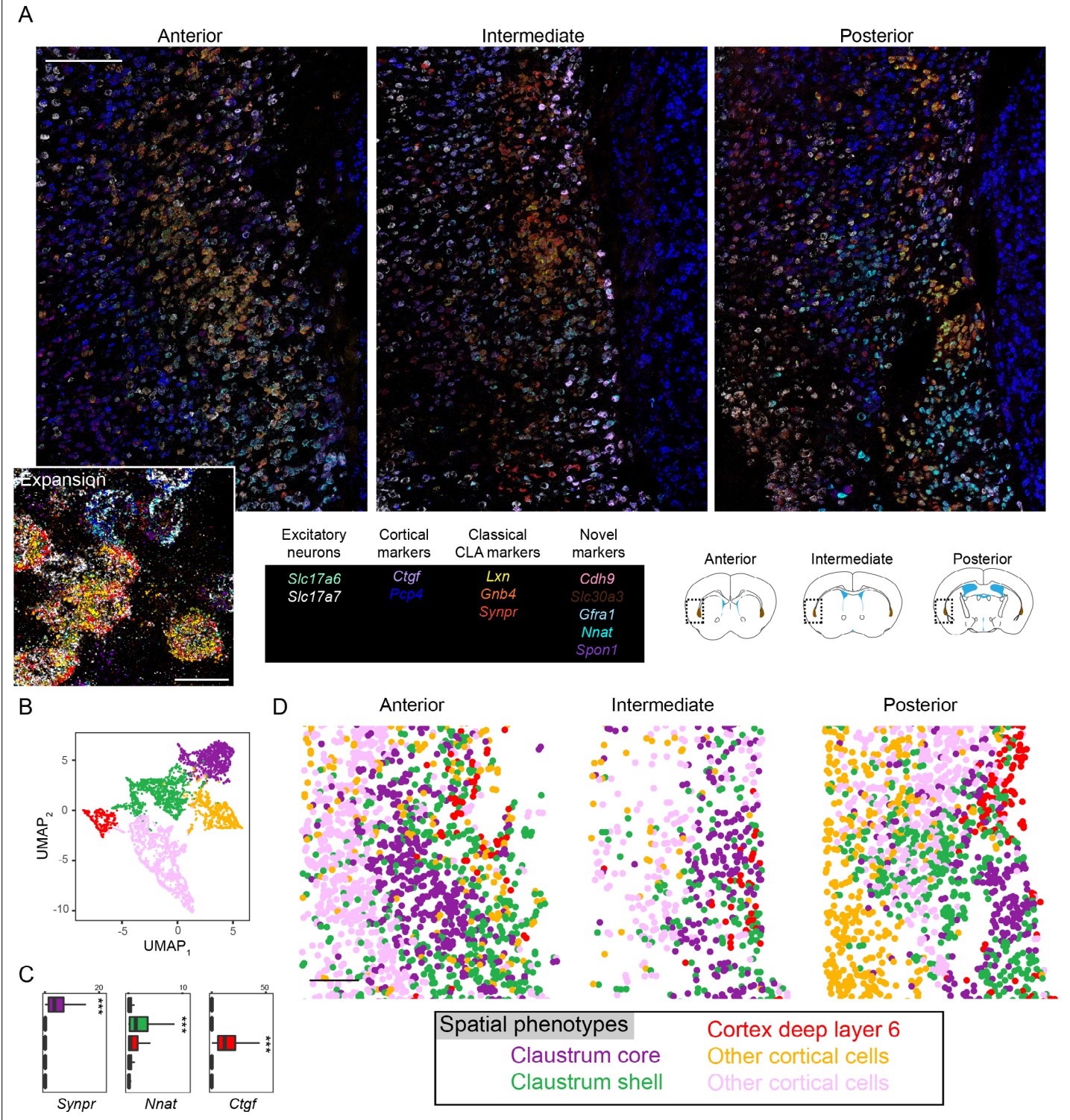

**Figure 2.** Multiplexed fluorescent in situ hybridization analysis of the claustrum. (**A**) Overview of anterior (left), intermediate (middle), and posterior (right) sections of the claustrum. Inset shows expansion of anterior section. Probe list provided at bottom middle, with atlas schematics denoting coronal section locations at bottom right, as well as imaged regions and claustrum definition of atlas (brown). Scale bars: overview: 200 µm; expansion: 20 µm. Atlas schematic adapted from *Franklin and Paxinos, 2013*. (**B**) UMAP-based nonlinear dimensionality reduction for *Slc17a7*-expressing cells (putative excitatory neurons) segmented from (**A**) and colored according to Leiden cluster identity. (**C**) Expression of example marker genes for core claustrum (*Synpr*), shell claustrum (*Nnat*), and layer 6 neurons (*Ctgf*). (**D**) Excitatory neurons from (**B**) plotted in spatial coordinates. Purple and green clusters respectively occupy the claustrum core and shell. Red neurons occupy deep layer 6 cortex, whereas yellow and pink clusters occupy other cortical

*Figure 2 continued on next page*

*Figure 2 continued*

regions. Scale bar: 200 μm.

The online version of this article includes the following figure supplement(s) for figure 2:

**Figure supplement 1.** scRNA-seq profiles of multiplexed fluorescent in situ hybridization-targeted genes.

**Figure supplement 2.** Representative expansions of RNA signals detected via multiplexed fluorescent in situ hybridization (mFISH).

**Figure supplement 3.** Expression of core, shell, and layer 6 marker genes across the anterior-posterior axis.

**Figure supplement 4.** Overview of cellular phenotyping across sections and mice.

**Figure supplement 5.** Comparison of gene expression of putative core and shell populations across scRNA-seq and multiplexed fluorescent in situ hybridization (mFISH).

associated with different long-range projection patterns (*Figure 3*). In previous projection mapping, it has been shown that the RSC and LEC reflect the two most spatially distinct core vs. shell projections (*Marriott et al., 2020*), and thus it is likely that other claustral projections comprise more of a mosaicism of core and shell transcriptomic phenotypes. Ultimately, this suggests a claustral organizational scheme wherein discretely separate transcriptomic subtypes are biased – but not wholly separable – according to long-range projection targets (*Cembrowski and Menon, 2018a*).

As the relationship between the claustrum and the nearby deep insular cortex and dorsal endopiriform cortex is often debated (*Bruguier et al., 2020*; *Marriott et al., 2020*; *Mathur, 2014*; *Mathur et al., 2009*; *Smith et al., 2019*; *Watakabe et al., 2012*; *Watson et al., 2017*; *Zingg et al., 2020*), it is important to discuss our work in the context of these adjacent regions. Relative to the deep insular cortex, our work here shows a general lack of deep cortical markers for two distinct neuron types, as well as enrichment of genes in these types that are not typically associated with deep cortex (*Figure 1C*, *Figure 1—figure supplements 3 and 4*). In conjunction with both of these neuron types showing enrichment of some claustrum marker genes relative to cortical neurons (e.g., *Gnb4*: *Figure 1C*), we interpret these transcriptomic cell types as claustrum core and shell neuron subtypes. Relative to the dorsal endopiriform cortex, our scRNA-seq and smFISH validation focused on the atlas-defined spatial extent of the claustrum; thus, future work targeting the dorsal endopiriform cortex will be needed to examine the transcriptomic relationship between these two regions.

Collectively, our results will allow subtype-specific claustral function to be assayed in future experiments by leveraging either marker genes or projection pathways (*Cembrowski, 2019*). Thus, our findings here will help to guide and inform observational and interventional experiments, and bridge claustrum cell-type identity, structure, and function. To facilitate such experiments and interpretations, we have hosted our scRNA-seq data online in conjunction with analysis and visualization tools (http://scrnaseq.janelia.org/claustrum). This web portal will help to identify how specific genes, cells, and circuits mechanistically drive claustral function and behavior.

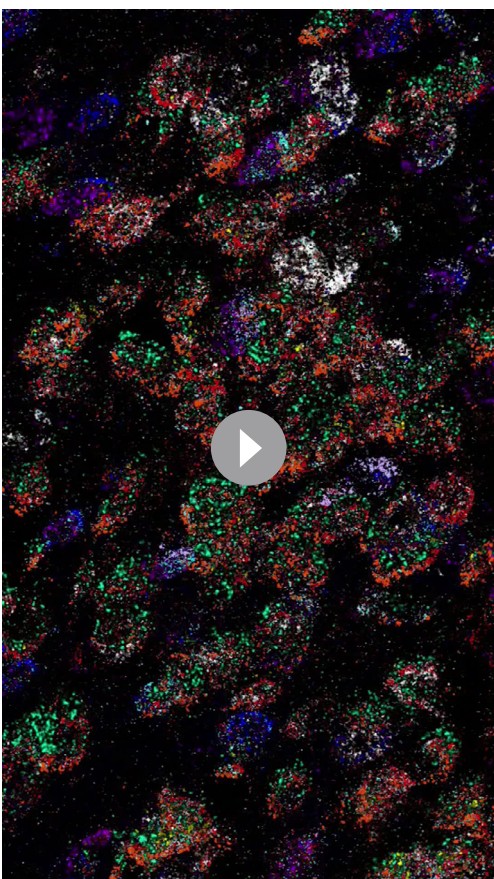

**Video 1.** Example multiplexed fluorescent in situ hybridization image across fine and coarse spatial scales.

https://elifesciences.org/articles/68967/figures#video1

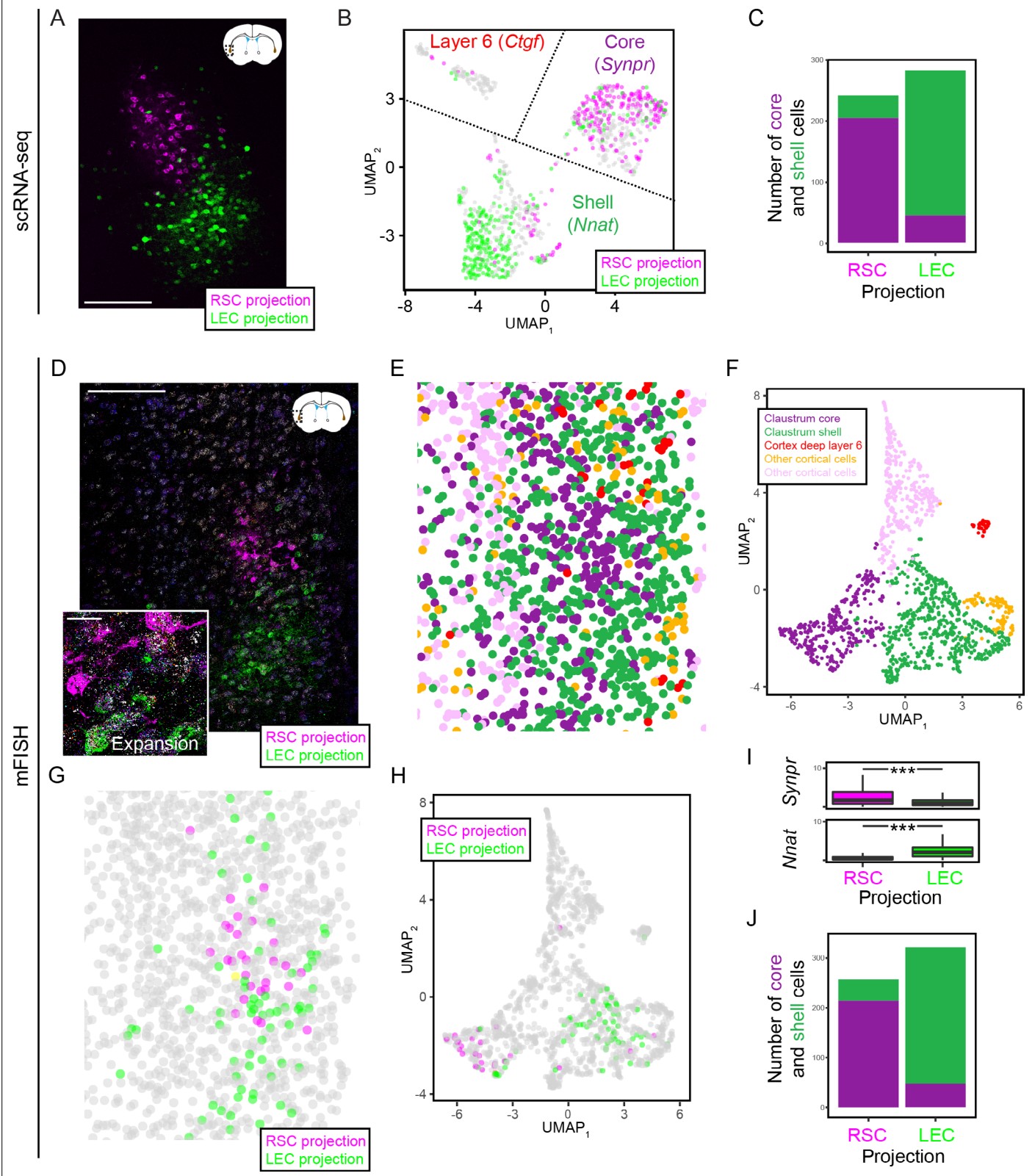

**Figure 3.** Claustrum transcriptomic subtypes are associated with different projections. (**A**) Projections to the retrosplenial cortex (RSC; magenta) and lateral entorhinal cortex (LEC; green) emanate from different spatial locations. Atlas schematic denotes coronal section location, adapted from *Franklin and Paxinos, 2013*. Scale bar: 200 μm. (**B**) Left: UMAP visualization of scRNA-seq claustrum transcriptomes, with coloring of individual cells corresponding to their associated projection. Labels denote cluster phenotypes and example marker genes. (**C**) Counts of RSC-projecting and

*Figure 3 continued on next page*

*Figure 3 continued*

LEC-projecting cells according to scRNA-seq core and shell phenotypes. (**D**) Representative multiplexed fluorescent in situ hybridization (mFISH) of intermediate claustrum section, including circuit mapping of long-range projections to the RSC (magenta) and LEC (green). Scale bars: overview: 200 µm; expansion: 20 µm. (**E**) Cellular segmentation and cluster identification based upon gene expression detected via mFISH, for section shown in (**D**). (**F**) UMAP dimensionality reduction of mFISH-characterized cells in (**D**), colored according to cluster identity as in (**E**). Putative phenotypes of clusters, based upon marker gene expression, are provided in inset. (**G**) Locations of neurons projecting to the RSC (magenta), LEC (green), or both (yellow), for section shown in (**D**). Scale bar: 200 µm. (**H**) As in (**G**), but with projections shown in UMAP embedding. (**I**) mFISH-derived expression of *Synpr* and *Nnat* in cells that project to either the RSC (magenta) or LEC (green). Results depict all projection-labeled cells across all sections and animals. (**J**) As in (**C**), but for mFISH core and shell phenotypes across all sections and animals.

© 2013, Franklin and Paxinos. Atlas schematic denotes coronal section location, adapted from *Franklin and Paxinos, 2013*. Further reproduction of this figure would need permission from the copyright holder.

The online version of this article includes the following figure supplement(s) for figure 3:

**Figure supplement 1.** Multiplexed fluorescent in situ hybridization (mFISH)-derived gene expression profiles for retrosplenial cortex (RSC)- and lateral entorhinal cortex (LEC)-projecting neurons.

# Materials and methods

## Key resources table

| Reagent type (species) or resource | Designation | Source or reference | Identifiers | Additional information |
| --- | --- | --- | --- | --- |
| Sequence-based reagent | *Cdh9* ISH probe | Advanced Cell Diagnostics | 443221-T1 | mFISH |
| Sequence-based reagent | *Ctgf* ISH probe | Advanced Cell Diagnostics | 314541-T2 | mFISH |
| Sequence-based reagent | *Slc17a6* ISH probe | Advanced Cell Diagnostics | 319171-T3 | mFISH |
| Sequence-based reagent | *Lxn* ISH probe | Advanced Cell Diagnostics | 585801-T4 | mFISH |
| Sequence-based reagent | *Slc30a3* ISH probe | Advanced Cell Diagnostics | 496291-T5 | mFISH |
| Sequence-based reagent | *Gfra1* ISH probe | Advanced Cell Diagnostics | 431781-T6 | mFISH |
| Sequence-based reagent | *Spon1* ISH probe | Advanced Cell Diagnostics | 492671-T7 | mFISH |
| Sequence-based reagent | *Gnb4* ISH probe | Advanced Cell Diagnostics | 460951-T8 | mFISH |
| Sequence-based reagent | *Nnat* ISH probe | Advanced Cell Diagnostics | 432631-T9 | mFISH |
| Sequence-based reagent | *Synpr* ISH probe | Advanced Cell Diagnostics | 500961-T10 | mFISH |
| Sequence-based reagent | *Pcp4* ISH probe | Advanced Cell Diagnostics | 402311-T11 | mFISH |
| Sequence-based reagent | *Slc17a7* ISH probe | Advanced Cell Diagnostics | 416631-T12 | mFISH |
| Software, algorithm | R | https://www.r-project.org | SCR_001905 | - |
| Software, algorithm | Seurat | https://satijalab.org/seurat/ | SCR_007322 | - |
| Software, algorithm | Fiji | https://imagej.net/Fiji | RRID:SCR_002285 | - |
| Other | rAAV2-retro-CAG-GFP | Janelia Viral Core | - | scRNA-seq |
| Other | pAAV-CAG-GFP | Addgene | RRID:Addgene_37825 | mFISH |
| Other | pAAV-CAG-tdTomato | Addgene | RRID:Addgene_59462 | mFISH |

All procedures were approved by the University of British Columbia Animal Care Committee (protocol A18-0267), the University of Alberta Health Science Laboratory Animal Services Animal Care and Use Committee (protocol AUP2711), and the Janelia Institutional Animal Care and Use Committee (protocol 17-159).

## Retrograde tracer injections

Mature C57BL/6 mice of either sex were used for injections, and randomly assigned retrograde injection locations and tracers. Mice were administered carprofen via ad libitum water 24 hr prior to surgery and for 72 hr after surgery to achieve a dose of 5 mg/kg. For surgery, mice were initially anesthetized using 4 % isoflurane and maintained with 1.0–2.5% isoflurane. Mice were secured in a stereotaxic frame, with body temperature maintained through an electric heating pad set at 37 °C, and lubricant was applied to eyes to prevent drying. Local anesthetic (bupivacaine) was applied locally

under the scalp, and an incision along midline was made to access bregma and all injection sites. Craniotomies were marked and manually drilled using a 400 µm dental drill bit according to stereotaxic coordinates. Pulled pipettes (10–20 µm in diameter) were back filled with mineral oil and loaded with virus or tracers. All injections were made using pressure injection, with 200 nL of retrograde tracer (*Tervo et al., 2016*) being injected. The skin was sutured after completing all injections and sealed. After allowing for sufficient time for retrograde labeling, mice were subsequently sacrificed for either histology, RNA sequencing, or mFISH processing, as described below.

## Single-cell RNA sequencing data acquisition and analysis

We used a manual capture approach to harvest cells from n = 4 mature male C57BL/6 mice. To facilitate microdissection of the claustrum, fluorescent tracers were used to delineate and grossly microdissect the claustrum from horizontal sections. In one animal, retrograde rAAV2-retro-CAG-GFP (*Tervo et al., 2016*) was injected into the anterior cingulate cortex to facilitate gross microdissection of the claustrum (*Jackson et al., 2018*), but not used to select for individual cells (i.e., GFP expression was used to microdissect the claustrum but cells were picked blind relative to GFP expression). In this animal, cells from separate anterior and posterior sections were obtained, allowing analysis of potential anterior vs. posterior differences in claustrum gene expression (*Figure 1—figure supplement 1B*). To build the projection-specific dataset, for the remaining three mice, green and red retrobeads were respectively injected into the LEC and RSC, with this labeling used for gross microdissection as well as to select a subset of projection-specific cells for RNA-seq.

In all cases, manual purification (*Hempel et al., 2007*) was used to capture cells in capillary needles in approximately 0.1–0.5 mL ACSF cocktail, placed into 8-well strips containing 3 µL of cell collection buffer (0.1 % Triton X-100, 0.2 U/µL RNAse inhibitor; Lucigen, Middleton, WI), and generally processed according to published methodology (*Cembrowski et al., 2018b*; *Schretter et al., 2020*). Specifically, each strip of cells was flash frozen on dry ice, then stored at –80 °C until cDNA synthesis. Cells were lysed by adding 1 µL lysis mix (50 mM Tris pH 8.0, 5 mM EDTA pH 8.0, 10 mM DTT, 1 % Tween-20, 1 % Triton X-100, 0.1 g/L Proteinase K [Roche], 2.5 mM dNTPs [Takara], and ERCC Mix 1 [Thermo Fisher] diluted to 1e-6) and 1 µL 10 µM barcoded RT primer (E3V6NEXT primer from *Soumillon et al., 2014*, modified to add a 1 bp spacer before the barcode, extending the barcode length from 6 bp to 8 bp, and designing the 384 barcodes to tolerate one mismatch error correction). The samples were incubated for 5 min at 50 °C to lyse the cells, followed by 20 min at 80 °C to inactivate the Proteinase K. Reverse transcription master mix (2 µL 5 X buffer; Thermo Fisher Scientific), 2 µL 5 M betaine (Sigma-Aldrich, St. Louis, MO), 0.2 µL 50 µM E5V6NEXT template switch oligo (Integrated DNA Technologies, Coralville, IA) (*Soumillon et al., 2014*), 0.1 µL 200 U/µL Maxima H- RT (Thermo Fisher Scientific), 0.1 µL 40 U/µL RNasin (Lucigen), and 0.6 µL nuclease-free water (Thermo Fisher Scientific) were added to the approximately 5.5 µL lysis reaction and incubated at 42 °C for 1.5 hr, followed by 10 min at 75 °C to inactivate reverse transcriptase. PCR was performed by adding 10 µL 2 X HiFi PCR mix (Kapa Biosystems) and 0.5 µl 60 µM SINGV6 primer with the following conditions: 98 °C for 3 min, 20 cycles of 98 °C for 20 s, 64 °C for 15 s, 72 °C for 4 min, with a final extension step of 5 min at 72 °C. Samples were pooled across the plate to yield approximately 2 mL pooled PCR reaction. From this, 500 µL was purified with 300 µL Ampure XP beads (0.6× ratio; Beckman Coulter, Brea, CA), washed twice with 75 % ethanol, and eluted in 20 µL nuclease-free water. The cDNA concentration was determined using Qubit High-Sensitivity DNA kit (Thermo Fisher Scientific).

13 plates were analyzed in total, with 600 pg cDNA from each plate of cells used in a modified Nextera XT (Illumina, San Diego, CA) library preparation with 5 µM P5NEXTPT5 primer (*Soumillon et al., 2014*). The resulting libraries were purified according to the Nextera XT protocol (0.6 × ratio) and quantified by qPCR using Kapa Library Quantification (Kapa Biosystems). Three plates were pooled on a NextSeq 550 mid-output flowcell with 26 bases in read 1, 8 bases for the i7 index, and 125 bases in read 2, and the remaining 10 plates were pooled on a NextSeq 550 high-output flowcell with 26 bases in read 1, 8 bases for the i7 index, and 50 bases for read 2. Alignment and count-based quantification of single-cell data was performed by removing adapters, tagging transcript reads to barcodes and UMIs, and aligned to the mouse genome via STAR (*Dobin et al., 2013*).

In total, 1112 cells were manually harvested and underwent sequencing. Of these initial 1112 cells, 27 putative non-neuronal cells were excluded due to low expression of *Snap25* (CPM < 0.001) and 74 additional cells were excluded due to low *Slc17a7* (CPM < 1e-10). The remaining 1011 cells exhibit 5.2

± 1.1 thousand expressed genes/cell from 142 ± 98 thousand reads/cell, mean ± SD. The relatively high abundance of excitatory neurons sampled owed both to the targeted approach for harvesting circuit-labeled cells, as well as the fact that excitatory neurons are relatively abundant relative to interneurons in the claustrum. No blinding or randomization was used for the construction or analysis of this dataset. No a priori sample size was determined for the number of animals or cells to use; note that previous methods have indicated that several hundred cells from a single animal are sufficient to resolve heterogeneity within excitatory neuronal cell types (*Cembrowski et al., 2018b*; *Cembrowski et al., 2018c*).

Computational analysis was performed in R (RRID:SCR_001905; *R Development Core Team, 2008*) using a combination of Seurat v3 (RRID:SCR_007322; *Satija et al., 2015*; *Stuart et al., 2019*) and custom scripts (*Cembrowski et al., 2018b*). To analyze our data, a Seurat object was created via *CreateSeuratObject*(min.cells = 3, min.features = 200), variable features identified via *FindVariableFeatures(selection.method='vst',nfeatures = 2000)* and scaled via *ScaleData*(). Data was processed via *RunPCA(), JackStraw(num.replicate = 100), RunTSNE(), FindNeighbors(), FindClusters(resolution = 0.1),* and *RunUMAP(reduction='pca')*, with 30 dimensions used throughout the analysis. This processed Seurat object was then used for downstream analysis. Subpopulation-specific enriched genes obeying $p_{ADJ}$ < 0.05 were obtained with Seurat via *FindMarkers*(), where $p_{ADJ}$ is the adjusted p-value from Seurat based on Bonferroni correction. Functionally relevant differentially expressed genes were obtained using *FindMarkers*(), allowing for both cluster-specific enriched and depleted genes obeying $p_{ADJ}$ < 0.05, and manually identified for functional relevance. Raw and processed scRNA-seq datasets have been deposited in the National Center for Biotechnology Information (NCBI) Gene Expression Omnibus under GEO: GSE149495.

To integrate and compare our scRNA-seq data to previously published data, we downloaded data from two previous studies that broadly sampled cortical cells in the mouse brain (*Saunders et al., 2018*; *Zeisel et al., 2018*). From *Saunders et al., 2018*, we downloaded frontal cortex data from F_GRCm38.81.P60Cortex_noRep5_FRONTALonly.raw.dge.txt.gz (from http://dropviz.org) and used a threshold of 16,000 transcripts/cell to extract 2877 total cells. After screening against cells that lacked *Snap25* and/or *Slc17a7* expression, 2842 putative excitatory neurons were retained for analysis (genes expressed/cell: 4.8 ± 0.6 thousand, mean ± SD; transcripts/cell: 16.5 ± 5.0 thousand, mean ± SD). We used a similar number of cells from *Zeisel et al., 2018*, obtained from l6_r4_telencephalon_projecting_excitatory_neurons.loom (from http://mousebrain.org/loomfiles_level_L6.html): 3151 cells were obtained using a threshold for 7500 transcripts/cell, with 3141 cells retained after requiring *Snap25* and *Slc17a7* expression (genes expressed/cell: 3.7 ± 0.4 thousand, mean ± SD; transcripts/cell: 9.6 ± 2.0 thousand, mean ± SD). Integration of these published datasets with our dataset was done in Seurat v3 (*Stuart et al., 2019*) by creating a Seurat object incorporating all datasets, and then using *SplitObject*() to split according to original dataset, allowing each dataset to independently undergo normalization and variable feature selection (handled identically to our data). Integration anchors were subsequently identified (via *FindIntegrationAnchors*()) and used for integration (via *IntegrateData*()), using 30 dimensions. From here, integrated data underwent scaling, dimensionality reduction, and clustering identically to the method used for our data, with clustering resolution = 2.5 to facilitate comparison between fine clusters associated with the claustrum shell. Statistical significance for adjusted p-values is denoted as follows: ns: $p \geq 0.05$; *p<0.05, **p<0.01, ***p<0.001.

## mFISH data acquisition and analysis

Custom probes for mFISH were purchased from Advanced Cell Diagnostics and were as follows: *Cdh9* (443221-T1), *Ctgf* (314541-T2), *Slc17a6* (319171-T3), *Lxn* (585801-T4), *Slc30a3* (496291-T5), *Gfra1* (431781-T6), *Spon1* (492671-T7), *Gnb4* (460951-T8), *Nnat* (432631-T9), *Synpr* (500961-T10), *Pcp4* (402311-T11), and *Slc17a7* (416631-T12). mFISH was generally performed as previously implemented (*Sullivan et al., 2020*). Briefly, mature male mice were randomly selected for mFISH and were deeply anesthetized with isoflurane and perfused with phosphate buffered saline (PBS) followed by 4 % paraformaldehyde (PFA) in PBS. Brains were dissected and post-fixed in 4 % PFA for 2–4 hr. Brain sections (20 μm) were made using a cryostat tissue slicer and mounted on glass slides. Slides were subsequently stored at –80 °C until use. For use, the tissue underwent pretreatment and antigen retrieval per the User Manual for Fixed Frozen Tissue (Advanced Cell Diagnostics). All 12 probes with unique tails (T1–T12) were hybridized to the tissue, amplified, and the tissue counterstained

with DAPI. Using cleavable fluorophores with unique tails (T1–T12), probes were visualized four at a time via an iterative process of imaging, decoverslipping, fluorophore cleaving, and adding the next four targeted fluorophores. mFISH performed on tissue with viral tracing was first counterstained with DAPI, coverslipped with ProLongGold antifade mounting medium, then imaged. The tissue was decoverslipped by soaking in 4× SSC. Following this, standard mFISH protocol was followed, with the antigen retrieval step quenching all endogenous viral fluorescent protein signal and DAPI signal.

mFISH images were acquired with a 63× objective on a SP8 Leica white light laser confocal microscope (Leica Microsystems, Concord, Ontario, Canada). Z-stacks were acquired with a step size of 0.45 μm for each imaging round. Final composite images are pseudocolored maximum intensity projections, including brightness adjustments applied to individual channels uniformly across the entire image, with channels opaquely overlaying one another ordered from highest to lowest expression.

Processing of mFISH images generally followed our previously published analysis pipeline (*Sullivan et al., 2020*) using Fiji (RRID:SCR_002285; *Schindelin et al., 2012*). Briefly, the DAPI signal from each round was used to rigidly register probe signals across rounds, followed by nonlinear elastic registration via bUnwarpJ (*Arganda-Carreras et al., 2010*) to accommodate any nonlinear tissue warping due to decoverslipping. The individual nuclei were then segmented and dilated by a factor of 5 μm to include the surrounding cytosol. The signal from each probe was then binarized by thresholding at the last 0.2–1% of the histogram tail, and then the number of pixels within regions of interest (ROIs) selected from segmentation was summed and normalized to the pixel area of the cell and multiplied by 100. This in effect corresponded to percent area covered (PAC) of the optical space of a cell.

A total of five mature male C57BL/6 mice, each with a relatively anterior, intermediate, and posterior section, were used for mFISH. Four mice had 200 nL retrograde viral injections into the RSC, two of which had additional retrograde viral injections into the LEC. For these injections, retrograde pAAV-CAG-GFP was a gift from Edward Boyden (Addgene viral prep # 37825-AAVrg; http://n2t.net/addgene:37825; RRID:Addgene_37825) and retrograde pAAV-CAG-tdTomato (codon diversified) was a gift from Edward Boyden (Addgene viral prep # 59462-AAVrg; http://n2t.net/addgene:59462; RRID:Addgene_59462). The last remaining mouse had no viral injections. Across these five animals, 33,155 cells in total were imaged. To facilitate analysis of excitatory neurons specifically, a threshold of one PAC of *Slc17a7* was required for each cell to be included in analysis, resulting in 18,957 total putative excitatory neurons being used for subsequent analysis. *Slc17a7* expression levels were used only for cellular phenotyping, and thus excluded from further analysis.

For analysis, UMAP dimensionality reduction (*McInnes et al., 2018*) was performed on within-cell normalized PAC values using the *umap* package (15 nearest neighbors, all other parameters default), and cells were clustered on a per-animal level using a Leiden community detection algorithm (*Levine et al., 2015*; *Traag et al., 2019*) via the *Monocle* package (*Cao et al., 2019*; *Qiu et al., 2017*; *Trapnell et al., 2014*). In general, setting the resolution parameter to a value that produced five clusters yielded strong agreement between UMAP dimensionality reduction and cluster assignments. Marker gene expression was used to assign phenotypes to cells comprising each cluster. For correlating projection targets with mFISH results, cells labeled with fluorescent retrograde tracers were manually identified (n = 739 total across n = 4 animals), done in a blinded fashion relative to mFISH analysis. A small minority of cells that projected to both the LEC and RSC (1.9%: n = 14/739, consistent with *Marriott et al., 2020*) were excluded when comparing properties of LEC- vs. RSC-projecting neurons. In general, box plots show distribution of gene expression contingent on cluster identity or projection target (hinges denote first and third quartiles, whiskers denote remaining data points up to at most 1.5 * interquartile range, outlier values beyond whiskers are not shown). Mann–Whitney U tests with a Bonferroni correction were used to identify differentially expressed genes, for either a given cluster relative to all other clusters, or in pairwise comparisons, as shown. Statistical significance for adjusted p-values is denoted as follows: ns: $p \geq 0.05$; *p<0.05, **p<0.01, ***p<0.001.

## Acknowledgements

MSC is supported by the University of British Columbia (Department of Cellular and Physiological Sciences, Djavad Mowafaghian Centre for Brain Health, and the Faculty of Medicine Research Office), the Natural Sciences and Engineering Research Council of Canada (RGPIN-2019-04507), the Canadian Institutes of Health Research (PJT-419798), and the Canadian Foundation for Innovation (John R Evans Leaders Fund 38369). KES is supported by a Royal Canadian Legion Masters Scholarship in

Veteran Health Research from the Canadian Institute for Military and Veteran Health Research. RMK is supported by a Fulbright U.S. Student Program Award. BM is supported by a studentship from the Neuroscience and Mental Health Institute. JJ is supported by the University of Alberta (Faculty of Medicine & Dentistry, and Department of Physiology), the Natural Sciences and Engineering Research Council of Canada (RGPIN-2018-05212), the Brain and Behavioral Research Foundation Young Investigator Grant, and Canadian Foundation for Innovation (John R Evans Leaders Fund). This work was supported by resources made available through the NeuroImaging and NeuroComputation Centre at the Djavad Mowafaghian Centre for Brain Health (RRID:SCR_019086). Collaboration between MSC and LW, JC and ALL was supported by the Janelia Visiting Scientist Program. We thank members of the Cembrowski lab for helpful discussions, and Jeffrey LeDue for insight and guidance in image acquisition.

## Additional information

### Funding

| Funder | Grant reference number | Author |
| --- | --- | --- |
| Natural Sciences and Engineering Research Council of Canada | RGPIN-2019-04507 | Mark S Cembrowski |
| Canadian Institutes of Health Research | PJT-419798 | Mark S Cembrowski |
| Canada Foundation for Innovation | John R. Evans Leaders Fund 38369 | Mark S Cembrowski |
| Natural Sciences and Engineering Research Council of Canada | RGPIN-2018-05212 | Jesse Jackson |
| Brain and Behavior Research Foundation | | Jesse Jackson |
| Canadian Institute for Military and Veteran Health Research | | Kaitlin E Sullivan |
| Natural Sciences and Engineering Research Council of Canada | | Kaitlin E Sullivan |
| Howard Hughes Medical Institute | Visiting Scientist Program | Mark S Cembrowski |
| Fulbright Association | US Student Program | Rennie M Kendrick |

The funders had no role in study design, data collection and interpretation, or the decision to submit the work for publication.

### Author contributions

Sarah R Erwin, Formal analysis, Investigation, Methodology, Validation, Writing - original draft, Writing – review and editing; Brianna N Bristow, Investigation, Methodology, Writing – review and editing; Kaitlin E Sullivan, Methodology, Software, Supervision; Rennie M Kendrick, Formal analysis; Brian Marriott, Methodology; Lihua Wang, Investigation, Methodology; Jody Clements, Resources, Software, Visualization; Andrew L Lemire, Investigation, Methodology, Supervision; Jesse Jackson, Conceptualization, Formal analysis, Methodology, Resources, Supervision, Writing - original draft, Writing – review and editing; Mark S Cembrowski, Conceptualization, Formal analysis, Funding acquisition, Investigation, Methodology, Resources, Supervision, Writing - original draft, Writing – review and editing

### Author ORCIDs

Sarah R Erwin http://orcid.org/0000-0001-8710-1848
Brianna N Bristow http://orcid.org/0000-0002-0411-916X

Kaitlin E Sullivan (iD) http://orcid.org/0000-0001-8043-7111
Andrew L Lemire (iD) http://orcid.org/0000-0002-0624-3789
Mark S Cembrowski (iD) http://orcid.org/0000-0001-8275-7362

## Ethics

All procedures were approved by the University of British Columbia Animal Care Committee (protocol A18-0267), the University of Alberta Health Science Laboratory Animal Services Animal Care and Use Committee (protocol AUP2711), and the Janelia Institutional Animal Care and Use Committee (protocol 17-159).

## Decision letter and Author response

Decision letter https://doi.org/10.7554/eLife.68967.sa1
Author response https://doi.org/10.7554/eLife.68967.sa2

## Additional files

### Supplementary files

- Supplementary file 1. List of core-enriched genes and enrichment properties.
- Supplementary file 2. List of shell-enriched genes and enrichment properties.
- Supplementary file 3. List of layer 6-enriched genes and enrichment properties.
- Transparent reporting form

### Data availability

Raw and processed scRNA-seq datasets have been deposited in the National Center for Biotechnology Information (NCBI) Gene Expression Omnibus under GEO: GSE149495.

The following dataset was generated:

| Author(s) | Year | Dataset title | Dataset URL | Database and Identifier |
|---|---|---|---|---|
| Cembrowski MS | 2020 | Core-shell organization in the mouse claustrum | https://www.ncbi.nlm.nih.gov/geo/query/acc.cgi?acc=GSE149495 | NCBI Gene Expression Omnibus, GSE149495 |

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
