## [Decision Letter]

**Acceptance summary:**

The authors used single-cell RNA sequencing to examine the molecular properties and axonal-projection targets of cell populations in the claustrum. The observed clusters of single cell RNA profiles reflect excitatory neuronal subtypes and their core-shell arrangements within the claustrum.

**Decision letter after peer review:**

[Editors’ note: the authors submitted for reconsideration following the decision after peer review. What follows is the decision letter after the first round of review.]

Thank you for submitting your work entitled "Spatially patterned excitatory neuron subtypes and circuits within the claustrum" for consideration by *eLife*. Your article has been reviewed by 3 peer reviewers, and the evaluation has been overseen by a Reviewing Editor and a Senior Editor. The following individual involved in review of your submission has agreed to reveal their identity: Arpiar Saunders (Reviewer #1).

Our decision has been reached after consultation between the reviewers. Based on these discussions and the individual reviews below, we regret to inform you that your work will not be considered further for publication in *eLife*.

Summary

The discussion among reviewers and the editors reached a clear consensus regarding the necessary experiments/analyses to strength the main claims regarding the relationship of the potential new cell types and the anatomical features relevant to claustrum function. We view the current interpretation of the results as limited by technical issues and the depth of the analysis. Closing that gap would require softening the claims of cell type discovery along with a more much thorough informatic analyses and likely follow up experiments (based on those re-analyses). *eLife* policy is not to consider a revision decision if we think that substantial additional experiments or analysis would be necessary to judge the acceptability of the manuscript and/or would require more than several months to complete.

*Reviewer #1:*

Erwin et al. address the relationship between molecular identity and projection anatomy of neurons in the mouse claustrum using single-cell RNAseq, in situ hybridizations and retrograde labeling. Such "mappings" are of interest to circuit neuroscience for both practical and theoretical reasons. In what ways does a neuron's molecular makeup determine its adult connectivity?

1. The molecular identity(ies) of the "Unknown" yellow cluster (Figures 1 and 3) still remain unknown. The Erwin manuscript implies that these cells make up a distinct cell type of claustrum shell, but the data presented supporting this conclusion are currently weak. The authors state that the "unknown" population is molecularly distinct from both (1) canonical claustrum neurons (Synpr+) and (2) other cortical neuron populations.

Due to the small number of cells sampled and the restriction of that sampling to the area of the claustrum, the first claim requires only that the canonical claustral neuron type be molecularly distinct to be supported. That there is a specialized claustrum neuron type (at the level of the entire cortical mantle or most likely the whole brain) is already quite clear: previous scRNAseq studies (Zeisel et al; Tasic et al; Saunders et al.) have demonstrated such a specialized RNA expression signature, which includes highly selective gene markers that are localized to the claustrum anatomically. The Erwin et al. data also support the claustrum-specific neuronal profile (as judged by heavy overlap of the selective markers from these other studies).

Using in situs, Erwin et al. refine the location of the Synpr+ cell type to claustrum "core". This is the major result of the paper.

The second claim – that the "unknown" cluster represents a 2nd distinct claustrum neuron type specific to the shell – is not well supported given the data at hand. The authors claim that because only 9.1% of Slc30a3+ cells are Pcp4+, the "unknown" cluster is not related to other excitatory cortical neuron types. However, Pcp4+ is not ubiquitously expressed in all Slc17a7+ neuron types (http://dropviz.org/?_state_id_=80467b8fc3286f0b). Moreover, quantitative variation in Pcp4 expression level could also lead to somewhat arbitrary thresholds for counting cells as positive, even with highly sensitive ACD RNA probes. By my eye, the expanded image in Figure 2D looks as if more than 10% of Slc30a3+ cells are Pcp4+ (if counting those cells with low Pcp4 expression; the authors should describe how the in situ data were quantified in the methods and if the representative images are Z-stack projections or single image planes).

More concerning is that when one considers the expression patterns of marker genes from the "unknown" cluster (Idb2, Mef2c, Necab1, Nnat, Nrn1, Olfm1, Spon1, Ramp1, Slc30a3, Syn2) across all of cortical cell types, those markers show very little cell-type-specific enrichment (http://dropviz.org/?_state_id_=4bb18bcc4f50d3cb). These data suggest to me that the "unknown" cluster could be a somewhat heterogenous collection that molecularly resemble other cortical excitatory neuron types. That their RNA profiles clustered together could be a property of their simply being dissimilar to the other more distinct cell types included in the analysis.

The authors can test their claim that "…the core-shell arrangement identified here was consistent with distinct subtypes claustral neurons, rather than reflecting spatial displacement of a cortical cell type" by jointly analyzing their own single cell libraries with much larger datasets describing molecular diversity of glutamatergic neurons across the cortical mantle (Zeisel et al; Tasic et al; Saunders et al). This type of analysis is not computationally prohibitive and could be performed on a decent laptop computer.

For example, if the Erwin and Saunders/Tasic/Zeisel data are co-clustered, and the authors find the cells from the "unknown" split into distinct excitatory clusters, that might suggest the "unknown" cluster is molecularly heterogenous and offer clues as to which cortical cell types functionally form the claustrum shell. Alternatively, the "unknown" cells might cluster together. This would allow an opportunity for cortex-wide differential expression testing to reveal selective markers for the "shell" subtype. (Importantly, the authors should pursue analytical methods (such as LIGER (Welch et al) or CCA (Butler et al.)) that allow their data to be integrated into other datasets without being burdened by platform/experiment specific clustering artifacts).

Non-clustering methods could also be used to attain similar insight.

To be clear, it would be an important observation and not diminish the study if the claustrum "shell" – a region defined anatomically by retrograde labeling – consists of a displaced cortical cell type or types.

More experiments are needed to address the issue of the molecular identity of the "shell" glutamatergic neurons; as it stands now, the claims in the paper are not strongly supported by the data presented.

2. Standard methods for single-cell 3' mRNA counting (e.g. Drop-seq, 10x, InDrop) routinely allow tens or hundreds of thousands of single cell RNA profiles to be sampled. It's unclear why the authors choose to pursue a more low-throughput method of whole-RNA sequencing but then not take advantage of mRNA isoform variation which their data uniquely enable? Single-cell RNAseq datasets describing the canonical claustrum neuron type (Synpr+/Nr4a2+) have been published with similar numbers of cells and transcripts/cell (Saunders et al. and Tasic et al; Tasic et al. also perform whole-RNA sequencing). Thus the Erwin et al. datasets is unlikely to reveal previously unappreciated heterogeneity with the canonical claustrum class.

3. Why were no interneurons sampled in the 274 cell dataset presented in Figure 1? If the authors were selecting particular dissociated cells by eye, they should describe by which criterion they choose individual cells. Could a claustrum excitatory neuron type with a small, irregular or interneuron like somato-dendritic morphology remain unsampled because of selection bias?

4. The authors should report more details for their molecular and sequencing methods. To identify how the single-cell RNA was processed, I had to track back through two publications (Cembrowski et al. 2018 -> Cembrowski et al. 2016) to find only "Total RNA was isolated from each sample, ERCC spike-in controls were added, and cDNA libraries were amplified from this material." Please provide detailed methods of library preparation.

5. The number of mice contributing to the presented experiments are sometimes very low, with some experiments n=2. The exact number of animals should be reported in the text of the results and bolstered where necessary.

The instance where I find this most concerning regards the data describing the relationship between molecular identities and LEC/RSC projection mapping; perhaps the most critical part of the manuscript.

I appreciate that the sequencing-based read out of projection anatomy is laborious and expensive. Why not simply combine two color retrograde labeling from LEC/RSC with in situs for shell/core markers? That would allow Erwin et al. to address variance in molecular identity/projection mapping across many mice. For experiments in which understanding biological variance is critical, the authors should also present their current (and any future) data, grouped by individual mouse (at least in the Supplement).

It's still very early days for circuit neuroscience to engender an understanding of how molecular and axonal anatomy of particular cell types "map". Critical in making strides in this effort are datasets that allow the across-animal variance of such mappings be accurately described so as to facilitate more nuanced conversations about such mappings. Erwin et all have an opportunity to make an important contribution if animal numbers are increased.

6. In Figure 3, do the ~15% of cells in each molecular cluster which do not obey the projection pattern of the majority other co-clustered cells exhibit a unique transcriptional signature? Again, this should be evaluated both within each mouse as well as across the projection-based cell populations as a whole

*Reviewer #2:*

My impression is that they may have a real finding regarding "shell" and "core" gene expression, but they don't do a convincing job showing it, especially with the FISH data. The study is kind of narrow in its scope for *ELife*, and should be further extended.

1. The core vs shell arrangement of gene expression is not so obvious in the FISH data provided in Figure 2D, yet this is a major claim made in the paper. In the given example, Slc30a3 appears medial to, and somewhat intermingled with Synpr, rather than forming a clear ring-like pattern as seen with the retrograde labeling example from LEC/RSC in Figure 3A. In fact, the experiment shown in Figure 3E provides another opportunity to convincingly present the core vs shell arrangement of Slc30a3 and Synpr as they relate to the projection-defined RSC population, however the provided images are cropped too closely to get a sense of the pattern. Perhaps provide an example cropped to the dimensions shown in Figure 3A.

2. The FISH signal for the CLA shell genes (Slc30a3 and Nnat) appears much weaker than that of Synpr and Pcp4 and some of the other examples provided. This hinders appreciation of the spatial arrangement of these genes around the CLA core and calls into question how reliable co-localization with other FISH probes could be identified. Perhaps potentially more robust probes or genes could be explored to strengthen the presentation of this data? For example, perhaps Spon1 (identified as a Cluster 2 gene in Figure 1F) or other unexplored genes, such as Sdk2 or Crym, that show striking shell-like patterns around the CLA might perform better in the FISH experiments? How well do they spatially colocalize with Slc30a3, which shows biased expression along the medial edge of CLA, but not as much around the lateral edge as seen with Nnat, Spon1, Sdk2, and Crym expression in Allen Institute's Mouse Brain ISH database (http://mouse.brain-map.org/)?

3. Pcp4 is used as a marker for L5/L6a/L6b cortical excitatory neurons and is claimed to be expressed ubiquitously among this population. However, it has been shown that in L6a, Pcp4 specifically labels corticothalamic-projecting neurons, but not L6 IT excitatory neurons, which are marked by a non-overlapping VGluT1^+^/CcK^+^ population (see Figure 8 in Watakabe et al., 2012; doi: 10.1002/cne.23160). Pcp4 is still a very good choice in these experiments for distinguishing between L6a CT and L6b cortical neurons and claustrum-related cells in the shell and core regions, perhaps just include the citation and modify the language in the text.

4. In Figure 3E, FISH is used to examine the colocalization of Synpr or Slc30a3 with retrogradely labeled RSC-projecting neurons in the CLA, however the same experiment is not performed with LEC-projecting cells and should be presented here. Injections of retrograde tracer in LEC indeed label cell bodies largely below and around the margins of the CLA "core," as shown in Figure 3A, however they may also label cortical neurons in the overlying insula, so it is unclear where the proposed boundaries of the CLA shell might be when looking at the LEC retrolabeling in Figure 3A. Demonstrating this retrograde labeling in conjunction with Slc30a3 and Pcp4 expression will help clear this up. Lastly, while difficult to address here, it will be important in the future to determine what fraction of the LEC retrolabeling shown below the CLA "core" in Figure 3A might actually comprise neurons belonging to the dorsal endopiriform nucleus (see Figure 8 of Watson et al., 2017; doi: 10.1002/cne.23981), which appears to also express genes for CLA "core" (Synpr), "shell" (Slc30a3, Nnat), and L6b (Ctgf) in ways that may complement or complicate the story unfolding for the CLA here. How the genes identified in this study potentially relate to the endopiriform complex (and even the dorsal aspect of CLA, thought to interact with motor cortex) may be worth mentioning in the discussion.

*Reviewer #3:*

The authors present a transcriptomic survey of cell type diversity in the mouse claustrum. They reveal that markers currently used to visualize cells in the core region of the claustrum may not target a second, potentially novel cell type in the claustrum shell. The authors additionally show that these two molecularly divergent cell types have distinct axonal projection targets in the cortex. While this is a potentially interesting finding, the data presented in the manuscript may not be sufficient to adequately describe the transcriptomic organization of the claustrum.

1) A large body of work, including the accompanying manuscript by the same authors, have described the claustrum as a topologically heterogeneous structure. The description of claustral cell types presented in the manuscript would be strengthened substantially if it addressed if and how gene expression and cell types vary spatially across the claustrum, as the authors have done in previous work with the subiculum, for example, and as the authors did anatomically in the related manuscript of Marriott et al. At a minimum, more detailed anatomical coordinates should be presented throughout for in situ/anatomical/sequencing data. It is unclear which parts of the claustrum are sampled and presented in the manuscript, making it difficult to discern if the results presented herein generalize across this structure or represent a snapshot of a single subregion, whose location is not specified

2) A major conclusion of the manuscript is that the claustrum contains two transcriptomically distinct cell types. However, the claustrum clearly contains other cell types that appear to be missing from scRNA-seq data. For example, 1) PV neurons are visible in the accompanying Marriott et al. and 2) PCP4+/CTGF- cells are present throughout (Figure 2, supplement 1). Presumably glia were also excluded, although there did not appear to be any mention of this. It is unclear why these types do not appear in scRNA-seq data, raising concerns about sampling biases and coverage.

3) This major finding of this study and that of Marriott et al., appear to overlap considerably – namely that the claustrum includes a 'shell' of cells with properties distinct from those in the 'core'. At the same time, the study appears to conflict in some ways with results in the accompanying Marriott et al. study. In particular, Marriott et al. show that only 'shell' neurons in the caudal aspect of the claustrum project to the EC, while this study implies that EC projections are a general feature of shell neurons, again raising questions about topology. This discrepancy makes one wonder if the rostral claustrum contains the same cell types transcriptomically.

4) A recent study (Wang, Xie, Gong, et al., bioRxiv) has described a large amount of heterogeneity in the projection patterns of Gnb4+ cells that presumably correspond to the authors' SYNPR population using single-neuron axonal reconstruction. These neurons appear to include projections to both the LEC and RSP. In addition, that study argued that the same transcriptomic population included cells both within and outside of the claustrum, further complicating the argument that a transcriptomic approach is the 'correct' method for defining its spatial borders. Those results should at least be discussed in the context of the present study.

[Editors’ note: further revisions were suggested prior to acceptance, as described below.]

Thank you for submitting your article "Spatially patterned excitatory neuron subtypes and circuits within the claustrum" for consideration by *eLife*. Your article has been reviewed by 3 peer reviewers, and the evaluation has been overseen by a Reviewing Editor and Gary Westbrook as the Senior Editor. The following individual involved in review of your submission has agreed to reveal their identity: Li I Zhang (Reviewer #3). The reviewers have discussed their reviews with one another, and the Reviewing Editor has drafted this to help you prepare a revised submission.

Essential revisions:

Please address the points raised by the reviewers, focusing on these issues:

1) Accurate articulation of the comparison between Erwin dataset and Saunders/Zeisel datasets, as requested by Reviewer #1.

2) Explain the apparent differences of Slc30a3 expression data across RNA-seq and different mFISH plots including Figure 2—figure supplement 2 and Figure 2—figure supplement 5.

3) Writing and presentation: Reviewers #1 and #2 have a number of suggestions for improving the readability, accuracy, figure labeling, and figure color scheme of the documentation. The reviewers and editors agree that addressing these concerns will improve the manuscript. In particular, Figure 2 is an important figure. The authors should include a better explanation of the data, including clusters displayed and the relationship with anatomical information. The delineation of 'core 'and 'shell' is the foundation of the work and Reviewer #2's concern concerning core/shell needs to be addressed. This critique also applies to Figure 3—figure supplement 2.

*Reviewer #1:*

Erwin et al's resubmission is improved by clarifying the methods and details of the experiments; inclusion of new, mFISH analyses with automated transcript quantification (including cells identified from retrograde labeling); and integration of their scRNA-seq datasets with those from larger, published studies. Their data make clear that there is a biased relationship between molecular identity and major axonal projection patterns for neurons in and around the claustrum which the authors describe as a "core" and "shell".

There are number of issues that could be addressed to strengthen the experiments, analyses and presentation of the manuscript.

1. The RNA profiles which make up the "unknown" cluster – which the authors name then analyze as the "shell" – have been sampled and described in other studies and appear, transcriptionally, to be more of a quantitative specialization within a larger neuron population rather than a qualitatively distinct cell type. I draw this inference from their integrated analysis (Figure 1—figure supplement 2). Contrast this to the cells from clusters 1 (canonical claustrum) and clusters 3 (Cortex L6) which appear to be distinct and discrete populations. Yet the way the manuscript is written suggests that the "shell" neurons are a categorically new and different population. I would suggest the authors change their description to better match the results presented in their integrated analysis.

2. The analysis in Figure 1—supplement 3 suffers from technical limitations and appears to be somewhat forcing the case that "shell" neurons they describe are distinct from those RNA profiles sampled in Saunders/Zeisel. Such comparisons of highly-granularized clusters can tend to identify expression "noise" rather than robust biological differences in RNA expression. Moreover, because each of the three studies used different chemistries and sequencing strategies to make scRNAseq libraries – and such chemistries have different DNA sequence-based biases in amplification and, moreover, were sequenced on different Illumina instruments – it seems in appropriate to compare expression levels of normalized RNAs across datasets in this manner. (I.e. that Nnat has higher expression in the Erwin dataset may simply reflect that Nnat cDNA was more likely to be present on the after Erwin-style library vs Dropseq or 10x V2 preparations).

3. Of the markers presented for Cluster 2, Slc30a3 appears to have the most selective expression. Yet the data in Figure 2—figure supplement 2 suggest no signal. Somewhat puzzlingly, the mFISH data for the same gene in Figure 2—figure supplement 5 suggest near equivalent levels of mRNA counts in "core" vs "shell" clusters. For many "shell" marker genes, the mFISH data does not appear to vary in "core" neurons to the degree that the scRNA-seq data suggest.

4. As a more general statement from 3 above, the violin plots used throughout the paper to show expression differences are hard to interpret, especially when severely flattened. The authors should perhaps consider a different plot type (maybe boxplots?) and to give the spacing to each plot so the differences they allude to can be evaluated. Are the violin plots on a log scale?

5. I find the mFISH analysis presented in Figure 2 confusing. Shouldn't panel C (the clustering results based on the mRNA counts of the 12 genes) be presented before panel B (the locations of cells in the cluster-categories). And critically, what are we to make of the yellow and pink cells in C? Their cell bodies appear to be located within the claustrum yet do they lack a homolog in the scRNAseq data?

6. The writing of the manuscript could be tightened to more carefully reflect what the authors intend. For example, the use of "circuit" – which I find somewhat ambiguous – might be better replaced by the words "cellular anatomy" or "axonal projection."

7. "Claustrum" is misspelled as "clautrum" in two instances.

*Reviewer #2:*

The revised manuscript of Erwin et al. has been improved though the addition of new experimental data and, as a result, my major comments have largely been addressed. While the results presented are better supported, the revised manuscript could be improved considerably by refinement in the way the data are presented and discussed, as detailed below.

*Reviewer #3:*

The manuscript is largely improved with additional data, especially the mFISH data. All my previous comments are addressed. I support its publication with *eLife*.

---

## [Author Response]

[Editors’ note: the authors resubmitted a revised version of the paper for consideration. What follows is the authors’ response to the first round of review.]

Reviewer #1:Erwin et al. address the relationship between molecular identity and projection anatomy of neurons in the mouse claustrum using single-cell RNAseq, in situ hybridizations and retrograde labeling. Such "mappings" are of interest to circuit neuroscience for both practical and theoretical reasons. In what ways does a neuron's molecular makeup determine its adult connectivity?1. The molecular identity(ies) of the "Unknown" yellow cluster (Figures 1 and 3) still remain unknown. The Erwin manuscript implies that these cells make up a distinct cell type of claustrum shell, but the data presented supporting this conclusion are currently weak. The authors state that the "unknown" population is molecularly distinct from both (1) canonical claustrum neurons (Synpr+) and (2) other cortical neuron populations.

We thank the reviewer for this feedback. Since this review, we have performed additional experiments and analysis to resolve this “Unknown” cluster in both cellular phenotype and spatial location. As described in detail below, this work has reinforced our original conclusion of this cluster being a claustral shell population. Of particular note, our revised manuscript now includes two wholly new figures (Figure 2, 3) devoted to this justification.

Due to the small number of cells sampled and the restriction of that sampling to the area of the claustrum, the first claim requires only that the canonical claustral neuron type be molecularly distinct to be supported. That there is a specialized claustrum neuron type (at the level of the entire cortical mantle or most likely the whole brain) is already quite clear: previous scRNAseq studies (Zeisel et al; Tasic et al; Saunders et al.) have demonstrated such a specialized RNA expression signature, which includes highly selective gene markers that are localized to the claustrum anatomically. The Erwin et al. data also support the claustrum-specific neuronal profile (as judged by heavy overlap of the selective markers from these other studies).

We agree that one of our clusters from our scRNA-seq dataset recapitulates a known claustral cell type, and in our revised manuscript, explicitly demonstrate this by integrating our scRNAseq dataset with previously published datasets (new Figure 1—figure supplements 2,3).

Using in situs, Erwin et al. refine the location of the Synpr+ cell type to claustrum "core". This is the major result of the paper.

In our revised manuscript, we now have greatly expanded our in situ hybridization experiments and analysis. In particular, using a multiplexed approach to register up to 14 channels from the same tissue section, we now simultaneously map the spatial organization of the claustrum core, shell, cortical cells, and their respective circuits (new Figures 2,3).

The second claim – that the "unknown" cluster represents a 2nd distinct claustrum neuron type specific to the shell – is not well supported given the data at hand. The authors claim that because only 9.1% of Slc30a3+ cells are Pcp4+, the "unknown" cluster is not related to other excitatory cortical neuron types. However, Pcp4+ is not ubiquitously expressed in all Slc17a7+ neuron types (http://dropviz.org/?_state_id_=80467b8fc3286f0b). Moreover, quantitative variation in Pcp4 expression level could also lead to somewhat arbitrary thresholds for counting cells as positive, even with highly sensitive ACD RNA probes. By my eye, the expanded image in Figure 2D looks as if more than 10% of Slc30a3+ cells are Pcp4+ (if counting those cells with low Pcp4 expression; the authors should describe how the in situ data were quantified in the methods and if the representative images are Z-stack projections or single image planes).

To avoid any potential biases associated with cell counting, as well as facilitate interpretation of our expanded 12- and 14-channel multiplexed in situ data, in our revised manuscript all quantification is now performed computationally (rather than manually in our previous manuscript). Our methods section is updated to describe this new computational analysis.

More concerning is that when one considers the expression patterns of marker genes from the "unknown" cluster (Idb2, Mef2c, Necab1, Nnat, Nrn1, Olfm1, Spon1, Ramp1, Slc30a3, Syn2) across all of cortical cell types, those markers show very little cell-type-specific enrichment (http://dropviz.org/?_state_id_=4bb18bcc4f50d3cb). These data suggest to me that the "unknown" cluster could be a somewhat heterogenous collection that molecularly resemble other cortical excitatory neuron types. That their RNA profiles clustered together could be a property of their simply being dissimilar to the other more distinct cell types included in the analysis.

In our original manuscript, the unknown cluster exhibited relatively small within-cluster variability, suggestive of a relatively homogeneous population. In our revised manuscript, we explicitly demonstrate this by integrating our single-cell RNA-seq dataset with previously published datasets, illustrating that this unknown population still tightly clusters within a much broader cell-type landscape (new Figure 1—figure supplement 2).

The authors can test their claim that "…the core-shell arrangement identified here was consistent with distinct subtypes claustral neurons, rather than reflecting spatial displacement of a cortical cell type" by jointly analyzing their own single cell libraries with much larger datasets describing molecular diversity of glutamatergic neurons across the cortical mantle (Zeisel et al; Tasic et al; Saunders et al). This type of analysis is not computationally prohibitive and could be performed on a decent laptop computer.

We thank the reviewer for this suggestion, and we now have implemented this integration in our revised manuscript (new Figure 1—figure supplement 2). Importantly, such analysis indicates that the “unknown” cluster is indeed a specific and relatively homogeneous cell type, and also has molecular features that distinguish it from published cortical cell types (new Figure 1 —figure supplement 3).

For example, if the Erwin and Saunders/Tasic/Zeisel data are co-clustered, and the authors find the cells from the "unknown" split into distinct excitatory clusters, that might suggest the "unknown" cluster is molecularly heterogenous and offer clues as to which cortical cell types functionally form the claustrum shell. Alternatively, the "unknown" cells might cluster together. This would allow an opportunity for cortex-wide differential expression testing to reveal selective markers for the "shell" subtype. (Importantly, the authors should pursue analytical methods (such as LIGER (Welch et al) or CCA (Butler et al.)) that allow their data to be integrated into other datasets without being burdened by platform/experiment specific clustering artifacts).

As requested, we use Seurat v3 data integration approaches to combine our scRNA-seq data with previous published datasets. This approach leverages an anchoring analytical method that circumvents platform/experiment-specific clustering artifacts, and we describe this new computational pipeline within our revised manuscript. With this approach, we find that our “Unknown” cluster encompasses a largely monolithic subtype (new Figure 1—figure supplement 2). This organization allowed us to identify selective markers for the shell type, as requested (new Figure 1—figure supplement 3).

Non-clustering methods could also be used to attain similar insight.

We now include UMAP embedding (new Figure 1—figure supplement 2) that allows visualization and comparison of datasets without clustering.

To be clear, it would be an important observation and not diminish the study if the claustrum "shell" – a region defined anatomically by retrograde labeling – consists of a displaced cortical cell type or types.

We agree, and appreciate the requests from the reviewer to further phenotype and understand our “Unknown” cluster.

More experiments are needed to address the issue of the molecular identity of the "shell" glutamatergic neurons; as it stands now, the claims in the paper are not strongly supported by the data presented.

We hope that, with our expanded analysis of scRNA-seq data (revised Figure 1) and complementary new experiments involved multiplexed FISH (new main Figures 2, 3 and associated 9 new supplemental figures), we have now illustrated the existence of shell claustral neurons.

2. Standard methods for single-cell 3' mRNA counting (e.g. Drop-seq, 10x, InDrop) routinely allow tens or hundreds of thousands of single cell RNA profiles to be sampled. It's unclear why the authors choose to pursue a more low-throughput method of whole-RNA sequencing but then not take advantage of mRNA isoform variation which their data uniquely enable? Single-cell RNAseq datasets describing the canonical claustrum neuron type (Synpr+/Nr4a2+) have been published with similar numbers of cells and transcripts/cell (Saunders et al. and Tasic et al; Tasic et al. also perform whole-RNA sequencing). Thus the Erwin et al. datasets is unlikely to reveal previously unappreciated heterogeneity with the canonical claustrum class.

We agree that mRNA isoform variation is an interesting are of research in general, but in the context of our study here, mRNA isoform variation is challenging in both scRNA-seq statistical inference as well as mFISH validation of scRNA-seq predictions. As our manuscript focuses on identifying subtypes of claustral neurons, which in principle and practice can be done at the relatively well-powered gene level, we elected to not perform any isoform-level analysis. We hope the reviewer finds this gene-level scope sufficient, especially as we believe our revised manuscript provides evidence of previously unappreciated heterogeneity within the claustrum.

3. Why were no interneurons sampled in the 274 cell dataset presented in Figure 1? If the authors were selecting particular dissociated cells by eye, they should describe by which criterion they choose individual cells. Could a claustrum excitatory neuron type with a small, irregular or interneuron like somato-dendritic morphology remain unsampled because of selection bias?

The cells harvested in our 274 cell dataset were chosen without selection criteria, and as such, morphology-based selection bias does not account for our relative enrichment of excitatory neurons. Rather, the relative enrichment of excitatory neurons in this dataset likely reflects the abundance of excitatory neurons relative to interneurons in the claustrum. We now include these details in our methods section.

We note that all of our scRNA-seq analysis in our revised manuscript now pools this dataset with our projection-level datasets (i.e., all analysis includes pooled data from n = 4 mice).

4. The authors should report more details for their molecular and sequencing methods. To identify how the single-cell RNA was processed, I had to track back through two publications (Cembrowski et al. 2018 → Cembrowski et al. 2016) to find only "Total RNA was isolated from each sample, ERCC spike-in controls were added, and cDNA libraries were amplified from this material." Please provide detailed methods of library preparation.

We now provide much greater detail throughout our methods, including library preparation.

5. The number of mice contributing to the presented experiments are sometimes very low, with some experiments n=2. The exact number of animals should be reported in the text of the results and bolstered where necessary.

We now include the exact number of animals in the text, and moreover, also provide demonstration of minimal animal-to-animal variability in our revised manuscript (scRNA-seq: new Figure 1—figure supplement 1C; mFISH: new Figure 2—figure supplement 4).

The instance where I find this most concerning regards the data describing the relationship between molecular identities and LEC/RSC projection mapping; perhaps the most critical part of the manuscript.I appreciate that the sequencing-based read out of projection anatomy is laborious and expensive. Why not simply combine two color retrograde labeling from LEC/RSC with in situs for shell/core markers? That would allow Erwin et al. to address variance in molecular identity/projection mapping across many mice.

We thank the reviewer for this suggestion, and in our revised manuscript, have expanded the number of animals, gene targets, and projection targets for relating molecular phenotype with circuit wiring. This has allowed for comprehensive phenotyping of LEC/RSC projections, and recapitulates and extends the results of our original manuscript (new Figure 3).

For experiments in which understanding biological variance is critical, the authors should also present their current (and any future) data, grouped by individual mouse (at least in the Supplement).

We now include our analysis on a per-replicate basis, both for scRNA-seq (Figure 1—figure supplement 1C) and mFISH (Figure 2—figure supplement 4).

It's still very early days for circuit neuroscience to engender an understanding of how molecular and axonal anatomy of particular cell types "map". Critical in making strides in this effort are datasets that allow the across-animal variance of such mappings be accurately described so as to facilitate more nuanced conversations about such mappings. Erwin et all have an opportunity to make an important contribution if animal numbers are increased.

Our revised manuscript is now expanded in the number of animals used for in situ hybridization experiments (n = 5 animals, cf. ≤2 animals/condition in our original manuscript). This expansion has enabled a dramatic increase in the number of cells examined (n = 18,957 putative excitatory neurons analyzed, cf. <1,000 in our original manuscript), as well as the genes/cell (12 genes/cell, cf. ≤3 in our original manuscript). We hope that this new data, along with our strengthened analysis and conclusions, will allow the reviewer to deem our work as an important contribution.

6. In Figure 3, do the ~15% of cells in each molecular cluster which do not obey the projection pattern of the majority other co-clustered cells exhibit a unique transcriptional signature? Again, this should be evaluated both within each mouse as well as across the projection-based cell populations as a whole

In general, we do not see dramatically different transcriptional signatures for cells that do not obey the typical projection pattern (e.g., new Figure 3F); however, we also note that due to the few number of cells in this category, our analysis is underpowered. Thus, if it pleases the reviewer, we would prefer to avoid speculating on potential transcriptional differences (or lack thereof) in cells that vary from the typical projection patterns.

Reviewer #2:My impression is that they may have a real finding regarding "shell" and "core" gene expression, but they don't do a convincing job showing it, especially with the FISH data. The study is kind of narrow in its scope for ELife, and should be further extended.

We thank the reviewer for the feedback. Since our initial submission, as we describe below, we have worked hard to extend the scope of our manuscript, including new 14-channel imaging results combining circuit mapping and multiplexed FISH.

1. The core vs shell arrangement of gene expression is not so obvious in the FISH data provided in Figure 2D, yet this is a major claim made in the paper. In the given example, Slc30a3 appears medial to, and somewhat intermingled with Synpr, rather than forming a clear ring-like pattern as seen with the retrograde labeling example from LEC/RSC in Figure 3A. In fact, the experiment shown in Figure 3E provides another opportunity to convincingly present the core vs shell arrangement of Slc30a3 and Synpr as they relate to the projection-defined RSC population, however the provided images are cropped too closely to get a sense of the pattern. Perhaps provide an example cropped to the dimensions shown in Figure 3A.

In our revised manuscript, using a much more powerful multiplexed FISH approach, we now provide much more expansive images that are also richer in gene expression information (new Figures 2, 3).

2. The FISH signal for the CLA shell genes (Slc30a3 and Nnat) appears much weaker than that of Synpr and Pcp4 and some of the other examples provided. This hinders appreciation of the spatial arrangement of these genes around the CLA core and calls into question how reliable co-localization with other FISH probes could be identified. Perhaps potentially more robust probes or genes could be explored to strengthen the presentation of this data?

As requested, in our revised manuscript we used a much larger number of genes (n = 12 genes per section in revised manuscript; cf. ≤ 3 in our original manuscript). Our findings using this expanded number of genes recapitulate the conclusions of our original manuscript.

For example, perhaps Spon1 (identified as a Cluster 2 gene in Figure 1F) or other unexplored genes, such as Sdk2 or Crym, that show striking shell-like patterns around the CLA might perform better in the FISH experiments? How well do they spatially colocalize with Slc30a3, which shows biased expression along the medial edge of CLA, but not as much around the lateral edge as seen with Nnat, Spon1, Sdk2, and Crym expression in Allen Institute's Mouse Brain ISH database (http://mouse.brain-map.org/)?

In our revised manuscript, from our scRNA-seq we identified 12 genes that in aggregate label a combination of core and shell claustrum cells, as well as other cell types (Figure 2—figure supplement 1). Given the combinatorial power of this gene set, we were able to use clustering analyses on mFISH data to identify core-shell organizations that did not rely on any individual gene per se*.*

3. Pcp4 is used as a marker for L5/L6a/L6b cortical excitatory neurons and is claimed to be expressed ubiquitously among this population. However, it has been shown that in L6a, Pcp4 specifically labels corticothalamic-projecting neurons, but not L6 IT excitatory neurons, which are marked by a non-overlapping VGluT1^+^/CcK^+^ population (see Figure 8 in Watakabe et al., 2012; doi: 10.1002/cne.23160). Pcp4 is still a very good choice in these experiments for distinguishing between L6a CT and L6b cortical neurons and claustrum-related cells in the shell and core regions, perhaps just include the citation and modify the language in the text.

We now include this citation and have modified the language in the text to illustrate this exception for Layer 6 intratelencephalic excitatory neurons.

4. In Figure 3E, FISH is used to examine the colocalization of Synpr or Slc30a3 with retrogradely labeled RSC-projecting neurons in the CLA, however the same experiment is not performed with LEC-projecting cells and should be presented here. Injections of retrograde tracer in LEC indeed label cell bodies largely below and around the margins of the CLA "core," as shown in Figure 3A, however they may also label cortical neurons in the overlying insula, so it is unclear where the proposed boundaries of the CLA shell might be when looking at the LEC retrolabeling in Figure 3A. Demonstrating this retrograde labeling in conjunction with Slc30a3 and Pcp4 expression will help clear this up.

We have now performed this experiment involving LEC-projecting cells (new Figure 3), and have found that our results from multiplexed FISH recapitulate the predicted results from our scRNA-seq data.

Lastly, while difficult to address here, it will be important in the future to determine what fraction of the LEC retrolabeling shown below the CLA "core" in Figure 3A might actually comprise neurons belonging to the dorsal endopiriform nucleus (see Figure 8 of Watson et al., 2017; doi: 10.1002/cne.23981), which appears to also express genes for CLA "core" (Synpr), "shell" (Slc30a3, Nnat), and L6b (Ctgf) in ways that may complement or complicate the story unfolding for the CLA here. How the genes identified in this study potentially relate to the endopiriform complex (and even the dorsal aspect of CLA, thought to interact with motor cortex) may be worth mentioning in the discussion.

We agree this is an important line of discussion. Given our expanded results, our revised manuscript is now at the character limit allowable for an *eLife* Short Report. If our revised manuscript is deemed potentially worthy of publication in *eLife*, we will request additional discussion space to address this point.

Reviewer #3:The authors present a transcriptomic survey of cell type diversity in the mouse claustrum. They reveal that markers currently used to visualize cells in the core region of the claustrum may not target a second, potentially novel cell type in the claustrum shell. The authors additionally show that these two molecularly divergent cell types have distinct axonal projection targets in the cortex. While this is a potentially interesting finding, the data presented in the manuscript may not be sufficient to adequately describe the transcriptomic organization of the claustrum.

We are glad that the reviewer found the findings of our previous manuscript potentially interesting. As described below, over the last 7 months we have dramatically expanded upon these original experiments and analysis. We hope our revised manuscript now sufficiently describes the transcriptomic organization of the claustrum.

1) A large body of work, including the accompanying manuscript by the same authors, have described the claustrum as a topologically heterogeneous structure. The description of claustral cell types presented in the manuscript would be strengthened substantially if it addressed if and how gene expression and cell types vary spatially across the claustrum, as the authors have done in previous work with the subiculum, for example, and as the authors did anatomically in the related manuscript of Marriott et al. At a minimum, more detailed anatomical coordinates should be presented throughout for in situ/anatomical/sequencing data. It is unclear which parts of the claustrum are sampled and presented in the manuscript, making it difficult to discern if the results presented herein generalize across this structure or represent a snapshot of a single subregion, whose location is not specified.

In our revised manuscript, we now include substantially more data and insight on location and variation in the anterior-posterior axis (e.g., scRNA-seq: new Figure 1—figure supplement 1; mFISH: new Figure 2, Figure 2—figure supplement 4, new Figure 3). This includes illustrating specific anatomical coordinates, as requested.

2) A major conclusion of the manuscript is that the claustrum contains two transcriptomically distinct cell types. However, the claustrum clearly contains other cell types that appear to be missing from scRNA-seq data. For example, 1) PV neurons are visible in the accompanying Marriott et al. and 2) PCP4+/CTGF- cells are present throughout (Figure 2, supplement 1). Presumably glia were also excluded, although there did not appear to be any mention of this. It is unclear why these types do not appear in scRNA-seq data, raising concerns about sampling biases and coverage.

We agree that our preparation methods – which select primarily for excitatory neurons due to overall abundance and further purification via circuit labeling – were poorly explained in our original submission. In our revised manuscript, we provide extended methods that explain the relative abundance of excitatory neurons observed in our dataset.

3) This major finding of this study and that of Marriott et al., appear to overlap considerably – namely that the claustrum includes a 'shell' of cells with properties distinct from those in the 'core'. At the same time, the study appears to conflict in some ways with results in the accompanying Marriott et al. study. In particular, Marriott et al. show that only 'shell' neurons in the caudal aspect of the claustrum project to the EC, while this study implies that EC projections are a general feature of shell neurons, again raising questions about topology. This discrepancy makes one wonder if the rostral claustrum contains the same cell types transcriptomically.

This point is well-taken, and in our revised manuscript, we now have now examined whether transcriptomic identity of claustrum neurons varies across the anterior-posterior (i.e., rostralcaudal) axis. Importantly, we do not find evidence of identity changes along this axis from either scRNA-seq data (new Figure 1—figure supplement 1C) or mFISH data (new Figure 2, new Figure 2—figure supplement 4).

4) A recent study (Wang, Xie, Gong, et al., bioRxiv) has described a large amount of heterogeneity in the projection patterns of Gnb4+ cells that presumably correspond to the authors' SYNPR population using single-neuron axonal reconstruction. These neurons appear to include projections to both the LEC and RSP.

We note that *Gnb4* is expressed in both core and shell subtypes in our study (e.g., Figure 1C), and thus, it is not unexpected that previous work showed projections to both LEC and RSC.

In addition, that study argued that the same transcriptomic population included cells both within and outside of the claustrum, further complicating the argument that a transcriptomic approach is the 'correct' method for defining its spatial borders. Those results should at least be discussed in the context of the present study.

We now cite this manuscript (Wang et al., bioRxiv 2017) along with the most recent version of this preprint (Peng et al., bioRxiv 2020) within our manuscript, and we furthermore agree that comparing our work to these preprints warrants further discussion. Given our expanded results, our revised manuscript is now at the character limit allowable for an *eLife* Short Report. If our revised manuscript is deemed potentially worthy of publication in *eLife*, we will request additional discussion space to address this point.

[Editors’ note: what follows is the authors’ response to the second round of review.]

Essential revisions:Please address the points raised by the reviewers, focusing on these issues:1) Accurate articulation of the comparison between Erwin dataset and Saunders/Zeisel datasets, as requested by Reviewer #1.

We have now performed a more in-depth comparison of our (Erwin) dataset to previous Saunders and Zeisel datasets, including new analysis and visualizations. This has resulted in a revised supplemental figure (Figure 1—figure supplement 4) and a new review figure (Author response image 1) . Our at-length response can be found after Comment 2 from Reviewer 1.

**Author response image 1. sa2fig1:** Similar expression of Pcp4 in the cortical cell cluster across datasets. Using cortical cells from our dataset (Erwin et al.), we identified the corresponding cluster from the integrated analysis including other published datasets (Saunders t al., Zeisel et al.). *Pcp4* expression is illustrated (via dimensionless units, corresponding to integrated scaled data), both for individual cells from each dataset (black dots), as well as associated violin plots on a per-dataset basis (coloured contours).

2) Explain the apparent differences of Slc30a3 expression data across RNA-seq and different mFISH plots including Figure 2—figure supplement 2 and Figure 2—figure supplement 5.

We have now updated our visualization of the data for both of these supplemental figures, and included additional statistical testing, both of which better capture the similarities between RNAseq and mFISH data. Our at-length response can be found after Comments 3 and 4 from Reviewer 1.

3) Writing and presentation: Reviewers #1 and #2 have a number of suggestions for improving the readability, accuracy, figure labeling, and figure color scheme of the documentation. The reviewers and editors agree that addressing these concerns will improve the manuscript. In particular, Figure 2 is an important figure. The authors should include a better explanation of the data, including clusters displayed and the relationship with anatomical information. The delineation of 'core 'and 'shell' is the foundation of the work and Reviewer #2's concern concerning core/shell needs to be addressed. This critique also applies to Figure 3—figure supplement 2.

We note that “Figure 3—figure supplement 2” is not a supplemental figure in our manuscript; given the context here we believe this comment refers to Figure 3—figure supplement 1.

As requested, in our revised manuscript we have now clarified our writing, as well as our presentation of associated figures. We have addressed all requested figure changes, which span the following revised main figures:

Figure 1

Figure 2 (see response to Comment 5 from Reviewer 1)

Figure 3 (see the following revised figure supplements):

Figure 1—figure supplement 1

Figure 1—figure supplement

Figure 1—figure supplement

Figure 1—figure supplement 4 (see response to Comment 2 from Reviewer 1)

Figure 2—figure supplement 1

Figure 2—figure supplement 2 (see response to Comments 3,4 from Reviewer 1)

Figure 2—figure supplement 5 (see response to Comments 3,4 from Reviewer 1)

Figure 3—figure supplement

Reviewer #1:Erwin et al.'s resubmission is improved by clarifying the methods and details of the experiments; inclusion of new, multiplexed FISH analyses with automated transcript quantification (including cells identified from retrograde labeling); and integration of their scRNA-seq datasets with those from larger, published studies. Their data make clear that there is a biased relationship between molecular identity and major axonal projection patterns for neurons in and around the claustrum which the authors describe as a "core" and "shell".There are number of issues that could be addressed to strengthen the experiments, analyses and presentation of the manuscript.1. The RNA profiles which make up the "unknown" cluster – which the authors name then analyze as the "shell" – have been sampled and described in other studies and appear, transcriptionally, to be more of a quantitative specialization within a larger neuron population rather than a qualitatively distinct cell type. I draw this inference from their integrated analysis (Figure 1—figure supplement 2). Contrast this to the cells from clusters 1 (canonical claustrum) and clusters 3 (Cortex L6) which appear to be distinct and discrete populations. Yet the way the manuscript is written suggests that the "shell" neurons are a categorically new and different population. I would suggest the authors change their description to better match the results presented in their integrated analysis.

We have now changed our description, emphasizing that our findings of the shell neurons are “specialized” and “distinguishable” relative to cortical neurons (revised Introduction, Discussion, and the following Results sections: “Comparison to previously published work” and “Two types of excitatory claustral neurons exist in a core-shell arrangement”).

2. The analysis in Figure 1—figure supplement 3 suffers from technical limitations and appears to be somewhat forcing the case that "shell" neurons they describe are distinct from those RNA profiles sampled in Saunders/Zeisel. Such comparisons of highly-granularized clusters can tend to identify expression "noise" rather than robust biological differences in RNA expression. Moreover, because each of the three studies used different chemistries and sequencing strategies to make scRNAseq libraries – and such chemistries have different DNA sequence-based biases in amplification and, moreover, were sequenced on different Illumina instruments – it seems in appropriate to compare expression levels of normalized RNAs across datasets in this manner. (I.e. that Nnat has higher expression in the Erwin dataset may simply reflect that Nnat cDNA was more likely to be present on the after Erwin-style library vs Dropseq or 10x V2 preparations).

We agree with the above general points raised by the reviewer, wherein different library preparation and sequencing strategies can introduce biases across scRNA-seq datasets. Nonetheless, we believe these concerns do not specifically apply to our integrated analysis examining either *Nnat* and *Pcp4*, and provide justification for this reasoning below.

For *Nnat*: Even though *Nnat-*expressing shell neurons are more abundant within our dataset, they are also found within both the Saunders and Zeisel datasets (see Figure 1—figure supplement 3B inset). Importantly, both Saunders and Zeisel datasets also contain shell neurons with higher *Nnat* expression than any cells within our dataset (see Figure 1—figure supplement 3D). Such results directly argue that reduction of *Nnat* in shell neurons reflects biological cell-type-specific differences, rather than technical differences across datasets.

For *Pcp4*: Shell neurons from our dataset exhibit reduced *Pcp4* expression relative to similar cells in Saunders and Zeisel datasets (Figure 1—figure supplement 3F), and notably our dataset was also largely absent of nearby *Pcp4-*expressing cells found in these previously published datasets (Figure 1—figure supplement 3B inset). Such results may be suggestive of technical differences leading to reduced *Pcp4* expression in our dataset, but also may reflect *bona fide* biological differences. To disambiguate between these two competing explanations, we leveraged the fact that *Pcp4* is a strong cortical marker, and thus examined *Pcp4* expression in our cortical cell type relative to Saunders and Zeisel datasets (cluster 3: Figure 1—figure supplement 2A). Here, we found that *Pcp4* was indeed similarly expressed across all datasets (Author response image 1). This result illustrates that decreased *Pcp4* expression is not broadly decreased across our dataset due to technical differences, but rather is decreased in a cell-type-specific fashion within our shell population.

Finally, in our revised manuscript we also now provide complementary ISH analysis that highlights why *Pcp4* is enriched in previous broad datasets, whereas *Nnat* is enriched in our targeted claustrum dataset (revised Figure 1—figure supplement 4). Specifically, *Pcp4* is broadly expressed across cortical layers but largely absent within the spatial extent of the claustrum, whereas *Nnat* is expressed around the claustrum but generally absent in other cortical regions. In combination with our expanded scRNA-seq analysis above, this work shows that our differential *Nnat* and *Pcp4* expression in our dataset emerges from our targeted approach to the claustrum, rather than reflecting technical differences across datasets.

3. Of the markers presented for Cluster 2, Slc30a3 appears to have the most selective expression.

We thank the reviewer for the below *Slc30a3-*related observations, and we note that there are several misperceptions here that we have now clarified in our revised manuscript.

Yet the data in Figure 2—figure supplement 2 suggest no signal.

We note that Figure 2—figure supplement 2 does indeed show *Slc30a3* signal. This was apparent in our original manuscript when looking at the expanded *Slc30a3* panel in (B), but may have been visually hard to discern due to our choice of dark brown for this gene. To address this, in our revised manuscript, we have converted gene-expression images within this supplemental figure to greyscale. This allows better comparisons across individual genes, and especially aids in visually identifying *Slc30a3* expression in particular (revised Figure 2—figure supplement 2).

Somewhat puzzlingly, the mFISH data for the same gene in Figure 2—figure supplement 5 suggest near equivalent levels of mRNA counts in "core" vs "shell" clusters.

We note that the supposed near-equivalent level of *Slc30a3* between core and shell clusters was not the case in our original manuscript, but this may have been visually hard to discern due to the use of violin plots to illustrate expression. To address this, in our revised manuscript, we have replaced violin plots with boxplots (as suggested by this reviewer, see comment 4 below). We furthermore now also include Mann-Whitney U tests, corrected for multiple comparisons, to statistically analyze differential expression between core and shell subtypes. With these revisions, it should now be more apparent that shell expression of *Slc30a3* clearly exceeds core expression in effect size and significance (revised Figure 2—figure supplement 5).

For many "shell" marker genes, the mFISH data does not appear to vary in "core" neurons to the degree that the scRNA-seq data suggest.

As discussed above, our revised manuscript incorporating boxplots and statistical tests more clearly illustrates that shell marker genes from mFISH recapitulate the relationships seen in scRNA-seq (see middle row in revised Figure 2—figure supplement 5).

4. As a more general statement from 3 above, the violin plots used throughout the paper to show expression differences are hard to interpret, especially when severely flattened. The authors should perhaps consider a different plot type (maybe boxplots?) and to give the spacing to each plot so the differences they allude to can be evaluated. Are the violin plots on a log scale?

For mFISH analysis, as well as when comparing mFISH to scRNA-seq, we now include boxplots on a linear scale as suggested. We note that when scRNA-seq data is presented in isolation in main figures (i.e., Figure 1), we have retained violin plots per standard scRNA-seq convention, plotted on a linear scale to facilitate comparisons with later analyses.

5. I find the mFISH analysis presented in Figure 2 confusing. Shouldn't panel C (the clustering results based on the mRNA counts of the 12 genes) be presented before panel B (the locations of cells in the cluster-categories).

The reviewer’s point is well-taken, and we have modified this figure accordingly.

And critically, what are we to make of the yellow and pink cells in C? Their cell bodies appear to be located within the claustrum yet do they lack a homolog in the scRNAseq data?

We thank the reviewer for bring this point to our attention. Motivated by this suggestion, we have revised our approach to clustering since our previous manuscript. In particular, we noted that our previous hierarchical clustering approach led to some discordancy with low-dimensional organization suggested by UMAP embedding, and thus sought to identify whether a different clustering approach could yield better agreement with dimensionality reduction.

In doing so, we identified Leiden clustering produces significantly better agreement with lowdimensional embedding (revised Figure 2, 3 and associated supplemental figures). These new cluster identities furthermore better capture the core-shell organization when viewed in a spatial context, and illustrate that cells contained within the atlas-defined claustrum predominantly derive from core, shell, and layer 6 neuronal identities.

6. The writing of the manuscript could be tightened to more carefully reflect what the authors intend. For example, the use of "circuit" – which I find somewhat ambiguous – might be better replaced by the words "cellular anatomy" or "axonal projection."

Broadly, we now use “projection” to refer to this organization in our revised manuscript, including changes in the title and abstract. We now only include “circuit” sparingly and only when the context is clear.

7. "Claustrum" is misspelled as "clautrum" in two instances.

Fixed.